# The actin nucleation promoting factor WASH facilitates clathrin-independent endocytosis of human papillomaviruses

Pia Brinkert [1], Lena Krebs[1], Pilar Samperio Ventayol [1], Lilo Greune [2], Carina Bannach[1], Cynthia Amakiri [1], Delia Bucher[3], Jana Kollasser[4], Petra Dersch [2], Steeve Boulant [3,5,7], Theresia E B Stradal[4] & Mario Schelhaas [1,6 ✉]

## Abstract

**Endocytosis is a fundamental cellular process facilitated by diverse mechanisms. Remarkably, several distinct clathrin-independent endocytic processes have been identified and characterized following virus uptake into cells. For some, however, mechanistic execution and biological function remain largely unclear. This includes an endocytic process exploited by human papillomavirus type 16 (HPV16). Using HPV16, we examine how vesicles are formed by combining systematic cellular perturbations with electron and video microscopy. Cargo uptake occurs by uncoated, inward-budding pits. Mechanistically, vesicle scission is facilitated by actin polymerization controlled through the actin nucleation-promoting factor WASH. While WASH typically functions in conjunction with the retromer complex on endosomes during retrograde trafficking, endocytic vesicle formation is largely independent of retromer itself and the heterodimeric membrane-bending SNX-BAR retromer adaptor, thereby uncovering a role of WASH in endocytosis in addition to its canonical role in intracellular membrane trafficking.**

**Keywords** Endocytosis; WASH; Actin; Nucleation Promoting Factor; Virus Entry
**Subject Categories** Membranes & Trafficking; Microbiology, Virology & Host Pathogen Interaction; Signal Transduction

## Introduction

Cells internalize extracellular ligands, fluids, plasma membrane lipids, and receptors by endocytosis. It is crucial for cellular and organismal homeostasis or responses by, for example, regulating signal transduction, intercellular communication, and neurotransmission (Schmid and Conner, 2003). Several endocytic mechanisms exist that are distinguished by diverse criteria. Morphologically, cargo is engulfed either by outward membrane protrusions (macropinocytosis, phagocytosis) or by inward budding pits (e.g., clathrin-mediated endocytosis (CME), glycosphingolipid-enriched endocytic carriers (GEEC)) (Heuser and Evans, 1980; Kirkham et al, 2005). The cellular machinery executing endocytic vacuole formation further defines the identity of endocytic pathways (Doherty and McMahon, 2009). For instance, macropinocytosis occurs by growth factor-induced, actin-driven protrusions that form large vacuoles upon protrusion backfolding with the help of the Bin1/Amphiphysin/Rvs (BAR) protein C-terminal binding protein 1 (CtBP1) (Liberali et al, 2008; Lim and Gleeson, 2011). Of the inward budding mechanisms, CME occurs by sequential recruitment of adapter proteins (AP) such as AP2 and the clathrin coat. Dynamin-2-mediated scission then leads to vacuole formation (McMahon and Boucrot, 2011; Merrifield and Kaksonen, 2014). In addition to the well-established pathways, a number of less well-understood endocytic processes have emerged. Limited information is available on which cellular processes they regulate, or on how vesicles are formed, as they are defined by their independence of classical endocytic regulators such as clathrin, caveolin, dynamin-2, or cholesterol (Doherty and McMahon, 2009). Interestingly, several of these mechanisms were identified by following the internalization of viruses such as lymphocytic choriomeningitis virus (LCMV), influenza A virus (IAV), and human papillomavirus type 16 (HPV16) (Mercer et al, 2010; Quirin et al, 2008; Rust et al, 2004; Schelhaas et al, 2012).

As intracellular parasites, viruses depend on host cells for their life cycle. This is particularly important during the initial phase of infection, termed entry, during which viruses deliver their genome to the site of replication within cells. As virus particles lack locomotive abilities, they strictly rely on cellular transport mechanisms to cross the plasma membrane, cytosol, or nuclear envelope. Following viral particles during entry thus allows

[1]Institute of Cellular Virology, ZMBE, University of Münster, Münster 48149, Germany. [2]Institute of Infectiology, ZMBE, University of Münster, Münster 48149, Germany. [3]Department of Infectious Diseases, Virology, Heidelberg University Hospital, Heidelberg 69120, Germany. [4]Department of Cell Biology, Helmholtz Centre for Infection Research, Braunschweig 38124, Germany. [5]German Cancer Research Center (DKFZ), Heidelberg 69120, Germany. [6]Cells in Motion Interfaculty Center, University of Münster, Münster 48149, Germany. [7]Present address: Department of Molecular Genetics and Microbiology, University of Florida, Gainesville, FL, USA. ✉E-mail: schelhaas@uni-muenster.de

researchers to analyze cellular transport mechanisms (Marsh and Helenius, 2006). For instance, studies on Semliki Forest virus (SFV) coined the field of endosomal trafficking, contributing to the identification of endosomes as pre-lysosomal structures (Helenius et al, 1983). Moreover, research on simian virus 40 (SV40) had a large impact on the understanding of caveolar endocytosis (Anderson et al, 1996; Pelkmans et al, 2001; Thorley et al, 2010). Our studies on HPV16 highlighted the existence of an endocytic mechanism independent of most known endocytic regulators (Schelhaas et al, 2012).

HPV16 is a small non-enveloped DNA virus and the leading cause of cervical cancer (de Sanjose et al, 2010; Mirabello et al, 2017). It initially infects basal keratinocytes of squamous mucosal epithelia, whereas completion of its life cycle requires suprabasal keratinocyte differentiation (Doorbar, 2005). Uptake of HPV16 into cells is slow and asynchronous, with quick individual virus internalization events occurring over a period of many hours after binding to cells (Becker et al, 2018; Schelhaas et al, 2012). Initial binding to heparan sulfate proteoglycans (HSPGs) induces crucial structural changes in the virus capsid that allow transfer to an internalization receptor complex to induce endocytosis (Bano et al, 2024; Becker et al, 2018; Cerqueira et al, 2013; Cerqueira et al, 2015; Feng et al, 2024; Giroglou et al, 2001; Richards et al, 2006; Selinka et al, 2007). While not understood in detail, the existing evidence supports a model in which the receptor complex constitutes specialized tetraspanin-enriched microdomains (TEMs) that resemble hemidesmosomes (HDs) consisting of the tetraspanin cluster of differentiation (CD) 151 (Fast et al, 2018; Scheffer et al, 2013), integrin α6 (Evander et al, 1997; Scheffer et al, 2013; Yoon et al, 2001) and epidermal growth factor receptor (EGFR) (Bannach et al, 2020; Schelhaas et al, 2012; Surviladze et al, 2012). After endocytosis, HPV16 traffics through the endosomal system (Bienkowska-Haba et al, 2012; Smith et al, 2008; Spoden et al, 2008). From endosomes, the virus is routed to the trans-Golgi network by retrograde transport (Day et al, 2013; Lipovsky et al, 2013) and reaches its site of replication, the nucleus, after nuclear envelope breakdown during mitosis (Aydin et al, 2014; Pyeon et al, 2009).

As indicated, HPV16 endocytosis is independent of a long list of prominent endocytic regulators, including clathrin, caveolin, dynamin, and cholesterol (Schelhaas et al, 2012). This endocytic mechanism depends on sodium/proton exchange, actin polymerization, signaling via EGFR, phosphatidylinositol-4,5-bisphosphate 3-kinase (PI3K), p21-activated kinase 1 (PAK1), protein kinase C (PKC), and Abelson tyrosine-protein kinase 2 (Abl2) (Bannach et al, 2020; Schelhaas et al, 2012). It remains largely elusive how these factors contribute to endocytic vesicle formation. We know, however, that EGFR and Abl2 regulate endocytosis induction and pit maturation, respectively (Bannach et al, 2020). As such, the requirements for this endocytic pathway are somewhat reminiscent of macropinocytosis, during which sodium/proton exchange regulates large actin-driven membrane protrusions to engulf bulk extracellular material (Doherty and McMahon, 2009; Mercer et al, 2010), and thus has been previously termed a macropinocytosis-like mechanism (Schelhaas et al, 2012). However, endocytosis results in small vesicles of about 60–100 nm diameter that are generated independently of cholesterol and the classical Rho-like GTPases cell division cycle 42 (Cdc42), rat sarcoma (Ras)-related C3 botulinum toxin substrate 1 (Rac1), and Ras homolog family

member 1 (RhoA) typically associated with macropinocytosis (Schelhaas et al, 2012).

Here, we addressed the mechanistic execution of vesicle formation using HPV16 pseudoviruses (PsV) as cargo. PsV comprise fully assembled virions that are immunologically indistinct from authentic viruses (Buck et al, 2005; Buck and Thompson, 2007; Doorbar et al, 2012), and that harbor a chromatinized reporter plasmid as a pseudogenome instead of the actual viral genome. We showed that cargo uptake occurred via uncoated vesicles severed from the plasma membrane by actin-related proteins (Arp) 2/3 complex-dependent branched actin polymerization. Interestingly, Arp2/3 complex activation occurred by the nucleation-promoting factor (NPF) Wiskott-Aldrich syndrome protein (WASP) and suppressor of cAMP receptor (Scar) homolog (WASH), a well-known regulator of endosomal cargo sorting, but not NPFs typically found at the plasma membrane. Distinct from its role at endosomes, WASH was not recruited to endocytic sites by the endosomal retromer complex or the membrane-bending sorting nexin (SNX) heterodimer. Moreover, the WASH function was independent of ubiquitylation at the established site K220, but required phosphorylation at Y141. Thus, our findings demonstrated a direct involvement of WASH in endocytosis distinct from its canonical role at endosomes.

## Results

### Endocytic vesicle formation occurs via uncoated, inward budding vesicles

Since the mechanism of how HPV16-containing endocytic vesicles are formed is mostly negative and thus ill-defined, we initially set out to elucidate their morphological itinerary. HPV16 was employed as trackable cargo in correlative light and electron microscopy (CLEM). To visualize different stages of endocytic vacuole formation and potential coats lining the forming vesicle as seen for clathrin-coated pits or caveolae, the signal of fluorescent virus particles imaged by confocal microscopy were correlated with structures on the cytosolic leaflet of plasma membrane sheets imaged by transmission EM of metal replicas (Bucher et al, 2018; Heuser and Evans, 1980; Sochacki et al, 2014). As inherent quality control, different stages of basally occurring clathrin-coated pit formation were readily observable at sites distinct from virus localizations (Fig. 1A). For HPV16, about 43% of fluorescent virus signals were associated with no obvious or clearly identifiable structure (Fig. 1B), likely representing at least in part virus binding to HSPGs prior to engagement of specialized TEMs and endocytosis induction. A considerable proportion (21%) was correlated with dense actin network patches (Fig. 1B). These may constitute anchoring structures for TEMs, either prior to or during induction of endocytosis (Ménager and Littman, 2016; Schelhaas et al, 2008b). Importantly, inward-budding structures at virus sites were readily observed and classified into stages of vesicle formation in analogy to CME (Fig. 1A,B). Designated as early stage, 10% of virions were associated with roughened plasma membrane patches representing initial curvature formation (Fig. 1A,B). Small invaginations (100–150 nm in diameter) were assigned as the intermediate stage, and represented pit maturation (10%). Fully rounded invaginations of 80–100 nm (16%) were classified as late-stage endocytic pits ready for scission (Fig. 1A,B). Notably, all invaginations were devoid of a

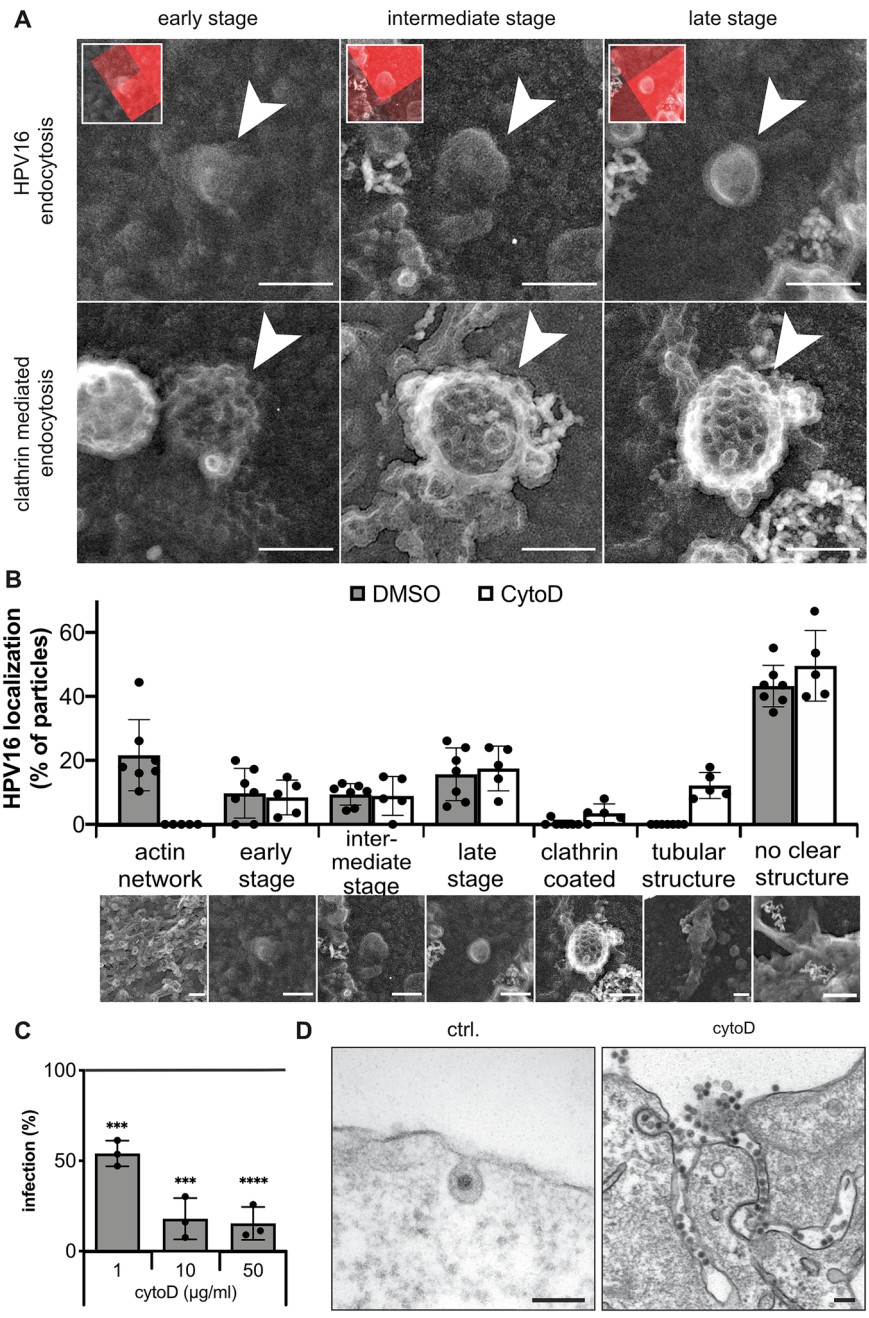

**Figure 1. HPV16 endocytosis is an uncoated, inward budding mechanism.**

(A) HaCaT cells were seeded on HPV16-AF568 bound to ECM, treated with 10 µg/ml cytoD or DMSO, and unroofed. Fluorescent virus particles were imaged by spinning disk microscopy and are depicted in the small insets. The virus signals were correlated with structures identified in transmission EM images of platinum/carbon replicas of unroofed membranes (HPV endocytosis). For comparison, unrelated, basally occurring clathrin-coated pit structures identified in transmission EM images of the same platinum/carbon replicas of unroofed membranes are shown (clathrin-mediated endocytosis). Arrowheads indicate endocytic pits. (B) Depicted is the percentage of particles associated with the indicated structure. Of note, about 43% of particles were not associated with a clearly identifiable structure (no clear structure). For clarity, EM images beneath the graph indicate a representative example of the quantified structure, which is in part reused from (A). (C) HeLa ATCC cells were infected with HPV16-GFP in the presence of cytoD. Infection was scored by flow cytometry and normalized to DMSO-treated controls. (n = 3 biological replicates). (D) HeLa ATCC cells treated with 10 µg/ml CytoD or left untreated (ctrl.) were infected with HPV16-GFP and processed for ultra-thin section transmission EM at 6 h p.i. Data information: For (B), shown is the mean ± variation of the cell membranes used in this experiment (DMSO: 134 viruses/7 membranes; cytoD: 101 viruses/5 membranes; constituting technical replicates). For (C), data are represented as mean ± SD for n = 3 biological replicates. Statistical significance was assessed by Student's t test (5 µg/ml: P = 0.0004, 10 µg/ml: P = 0.0002, 50 µg/ml: P ≤ 0.0001). All scale bars in this figure are 100 nm. Dots in bar graphs indicate the replicates.

discernible coat or other regular structures (Fig. 1A). Thus, cargo uptake in the unique endocytic pathway occurred by a stepwise, inward budding process, in which endocytic vacuoles were formed without observable contribution of a coat structure. Inferring from the stepwise formation of vesicles, different sets of cellular proteins would have to be recruited to function in a sequential manner thereby facilitating membrane invagination, pit dilation, constriction, and scission in analogy to CME.

In the studied mechanism, most typical endocytic regulators, including dynamin, are dispensable for vesicle formation. However, actin polymerization facilitates cargo uptake (Schelhaas et al, 2012). Expectedly, perturbation of actin assembly by cytochalasin D (cytoD) reduced HPV16 infection (Fig. 1C). To analyze how defective actin assembly impacts endocytic pit formation, we used CLEM of cytoD-treated cells. As expected, the actin network population was no longer detectable (Fig. 1B). However, all stages of membrane invaginations were present to a similar extend in cytoD- and untreated samples (Fig. 1B), indicating that endocytosis induction and pit formation were independent of actin polymerization. Notably, virus-correlated tubular structures devoid of an apparent coat newly appeared in cytoD-treated cells (Fig. 1B). In fact, these tubular membrane invaginations were filled with virus particles (Fig. 1D). Taken together, the formation of endocytic structures and the failure to separate cargo-filled tubules from the plasma membrane imply that endocytic vesicle scission but not induction and membrane invagination were facilitated by polymerizing actin filaments.

## Actin polymerization coincides with cargo internalization

Consistent with a role of actin polymerization aiding vesicle scission, the presence of actin was detectable close to constricted endocytic pits in immunogold labeling EM (Fig. 2A). To provide direct evidence for induction of actin polymerization during virus endocytosis, and to gain insights into how actin may facilitate vesicle scission, we employed live-cell TIRF microscopy (TIRF-M) to selectively illuminate the basal plasma membrane. Denoted by a rapid loss of virus signal, vesicles are formed within about 2 min, once HPV16 has been committed to uptake (Schelhaas et al, 2012). Here, cargo uptake into the cell interior correlated with an increase of filamentous actin, indicating actin polymerization at the time of scission (Fig. 2B,C; Movie EV1). Detailed analysis of intensity profiles revealed that the loss of virus signal took about 4 s (Fig. 2C,D; Movie EV1). While the onset of actin polymerization was somewhat variable, the increase in actin signals was always initiated prior to cargo internalization (about 10 s on average), peaked just before cargo uptake, and decreased thereafter (Fig. 2C,D; Movie EV1). The dynamics of actin polymerization resembled dynamin recruitment during scission in CME, starting about 20 s prior to loss of the clathrin signal from the plasma membrane (Fig. EV1A–C; Movie EV2) (Merrifield et al, 2002). Since dynamin is dispensable for cargo uptake in this pathway (Schelhaas et al, 2012; Spoden et al, 2008), actin likely replaced dynamin functionally as a scission factor. Moreover, the local, transient increase in filamentous actin indicated that polymerized actin did not merely serve as an anchor for other scission factors, but was specifically induced for and actively contributed to endocytic vesicle scission as for other endocytic mechanisms (Merrifield et al, 2002; Mooren et al, 2012; Pelkmans et al, 2002).

## Branched actin polymerization regulates cargo uptake

To understand the mechanism of actin polymerization for endocytic vesicle scission, we investigated the involvement of branched versus unbranched actin filaments. Formin-mediated unbranched actin polymerization was perturbed by SMIFH2, a small molecule inhibitor binding the formin homology 2 domain in formins (Rizvi et al, 2009). SMIFH2 treatment dose-dependently interfered with actin-dependent CME, as evidenced by a decrease in vesicular stomatitis virus (VSV) infection (Fig. 3A) (Cureton et al, 2009). In contrast, SMIFH2 did not affect HPV16 infection (Fig. 3A). Furthermore, HPV16 infection was largely unaffected by siRNA-mediated depletion of individual formins (Fig. EV2A). However, sequestering the Arp2/3 complex as key regulator of branched actin polymerization by overexpression of the Arp2/3 binding domain of WASP and WASP-family verprolin-homologous protein (WAVE) (Scar-WA) (Machesky and Insall, 1998) strongly reduced HPV16 infection compared to cells expressing a control lacking the Arp2/3 binding domain (Scar-W) (Fig. 3B). In line with a requirement of branched actin polymerization, RNA interference (RNAi) with Arp3 expression reduced infection by about 80% (Fig. 3C) compared to cells transfected with a non-targeting control (ctrl.). Similarly, vaccinia virus (VV) infection by macropinocytosis (Mazzon and Mercer, 2014; Mercer and Helenius, 2008) was reduced upon Arp3 depletion, whereas Semliki Forest virus (SFV) uptake by actin-independent CME (DeTulleo and Kirchhausen, 1998; Marsh and Helenius, 1980; Marsh et al, 1984) was even increased (Fig. 3C). Taken together, active branched but not unbranched actin polymerization was crucial.

To directly verify the impact on cargo uptake, a bona fide endocytosis assay was employed. For this, cells were infected with HPV16 covalently labeled with the pH-sensitive dye pHrodo, which gives rise to fluorescence only upon delivery to acidic endosomal organelles (Fig. 3D) (Becker et al, 2018; Samperio Ventayol and Schelhaas, 2015). Depletion of Arp3 reduced cargo delivery to endosomes by about 70% (Fig. 3D). Branched actin polymerization might aid intracellular trafficking of the virus. However, it is most likely essential for endocytosis, because virus particle translocation across the plasma membrane but not virus binding is abrogated by inhibition of all actin polymerization (Schelhaas et al, 2012).

## WASH is the major NPF in HPV16 infection

Since the Arp2/3 complex is inherently inactive, how is it activated for virus uptake? Several NPFs with distinct cellular functions and localizations regulate Arp2/3 complex activity (Rottner et al, 2010; Stradal and Scita, 2006). To identify which NPF acted here, we systematically employed RNAi against all known NPFs in combination with HPV16 infection in HeLa cells. Depletion of the most prominent actin regulators in endocytic processes, neuronal WASP (N-WASP) and WAVE isoforms 1 or 2 (Chadda et al, 2007; Qualmann and Kelly, 2000; Suetsugu et al, 2003), did not alter HPV16 infection (Fig. EV2B–D,G). Since N-WASP, WAVE1, and 2 were dispensable, a potential involvement of NPFs typically not found at the plasma membrane was assessed next. Neither depletion of junction mediating and regulatory protein, p53 cofactor (JMY), a regulator of DNA damage response and cell migration (Shikama et al, 1999; Zuchero et al, 2009), nor depletion of WASP homolog associated with actin, Golgi membranes, and

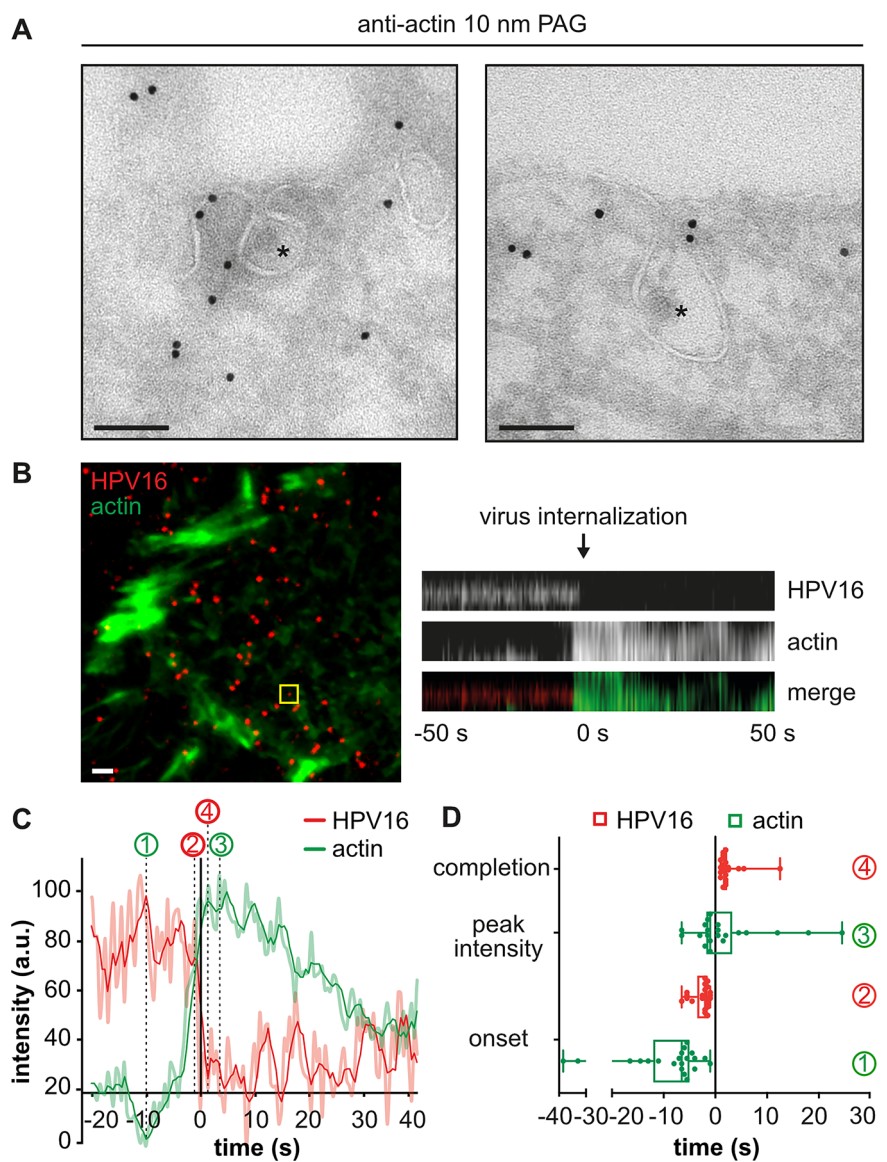

**Figure 2. Actin polymerization facilitates cargo internalization.**

(A) HeLa ATCC cells were infected with HPV16-GFP, subjected to immunogold labeling of actin on ultra-thin cryosections, and analyzed by transmission EM. Asterisks indicate HPV16 in endocytic pits. (B) HeLa ATCC cells were transfected with lifeact-EGFP, infected with HPV16-AF594, and imaged by live-cell TIRF-M at 1 h p.i. Movies were acquired with 0.5 Hz frame rate for 5 min. HPV16 entry events were identified manually after background subtraction and filtering. The yellow box indicates the virus entry event shown as a kymograph (right) and intensity profile in (C). (C) Plotted are the intensity profiles of HPV16 and lifeact (light red/green) as well as moving averages (intense red/green). Values are depicted relative to the half-time of virus loss ($t = 0$). The time points of the onset of actin polymerization (1) and of virus signal loss (2) from the cell surface, as well as of actin peak intensity (3) and of completion of virus uptake (4) were determined manually. (D) Time points were determined for 21 virus entry events (biological replicates) as indicated in (C), averaged, and are depicted as box plots. Data information: (A) Scale bars are 100 nm. (B) Scale bar is 2 µm. (D) Shown are 21 individual entry events from $n = 7$ independent experiments. Dots indicate the individual entry events. Box plots indicate the interquartile range (25th to 75th percentiles, box) and median, while the whiskers extend to the minimum and maximum values of all data points.

microtubules (WHAMM), which is involved in the secretory pathway (Campellone et al, 2008), impaired HPV16 infection (Fig. EV2E–G). However, depletion of WASH, a well-known actin regulator of endosomal cargo sorting (Derivery et al, 2009; Duleh and Welch, 2010; Linardopoulou et al, 2007), strongly reduced HPV16 infection in HeLa cells, whereas VV uptake by macropinocytosis was mostly unaffected (Figs. 4A and EV2G). In conclusion, the requirement of WASH as the NPF for HPV16

infection suggested that WASH, rather than NPFs typically found at the plasma membrane, acted in HPV16 uptake.

## WASH function during virus infection is distinct from its function at endosomes

To our knowledge, no evidence exists that WASH directly facilitates endocytic vesicle formation. However, since it was the

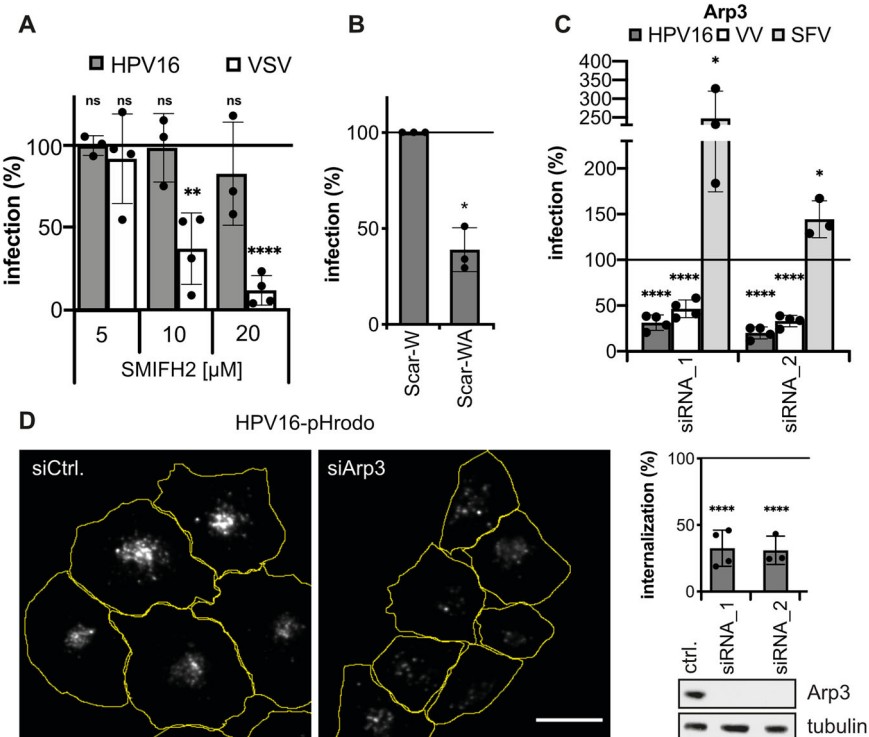

**Figure 3. Branched actin polymerization mediates endocytosis.**

(A) HeLa ATCC cells were infected with HPV16-GFP or VSV-GFP in the presence of the formin inhibitor SMIFH2. Infection was scored by flow cytometry at 48 h p.i., normalized to solvent-treated controls. (B) HeLa ATCC cells transfected with Scar-W-GFP or Scar-WA-GFP were infected with HPV16-RFP. Infection of transfected cells was analyzed by microscopy, normalized to Scar-W-GFP control cells. (C) HeLa Kyoto cells were depleted of Arp3 and infected with HPV16-GFP, VV-GFP, or SFV. Infection was scored by automated microscopy or flow cytometry, normalized to cells transfected with a non-targeting control siRNA (ctrl.). (D) Arp3 depletion was followed by infection with HPV16-pHrodo and live-cell spinning disk microscopy at 6 h p.i. Shown are average intensity projections of the HPV16-pHrodo signal with cell outlines (yellow), scale bar is 25 μm. Virus signal intensities per cell were measured using a CellProfiler pipeline and normalized to ctrl. Knockdown levels were analyzed by Western blotting against Arp3. Data information: In (A–D), data are represented as mean ± SD. Statistical significance was assessed by Student's $t$ test for all panels except (B), where a Welsh's $t$ test has been used. Significance is depicted as (*$P \leq 0.05$, **$P \leq 0.01$, ***$P \leq 0.001$, ****$P \leq 0.0001$, ns = not significant). For (A), data are from $n = 3$ (HPV16), or $n = 4$ (VSV) biological replicates. Significance values are HPV16 - 5 μM: $P = 0.9232$, 10 μM: $P = 0.8895$, 20 μM: $P = 0.3875$; VSV: 5 μM: $P = 0.5602$, 10 μM: $P = 0.0012$, 20 μM: $P \leq 0.0001$. For (B), data are depicted as the mean of $n = 3$ biological replicates ± SD. Statistical significance: $P = 0.0115$. For (C), data are from $n = 4$ (HPV16) or $n = 3$ (SFV) biological replicates and depicted as mean ± SD. Significance values are HPV16 & VV: siRNA 1&2: $P \leq 0.0001$; SFV: siRNA_1: $P = 0.0249$, siRNA_2: $P = 0.0188$. For (D), data are depicted as the mean of $n = 4$ (siRNA_1) and $n = 3$ (siRNA_2) biological replicates ± SD. Significance values are siRNA_1 & _2: $P \leq 0.0001$. Dots in bar graphs indicate the replicates.

only NPF important for HPV16 infection, we hypothesized a role of WASH in regulating actin polymerization for endocytosis. Consistently, silencing of WASH resulted in a strong decrease of cargo uptake and of cargo delivery to endosomes comparable to RNAi of Arp2/3 (Figs. 4B and EV3G, compare Fig. 3D). Unaffected HPV16 binding to WASH KO cells or upon RNAi of WASH suggested that recycling and binding receptor presentation were in principle intact (Fig. EV3E). Moreover, plasma membrane presentation of the internalization receptor candidates CD151 and EGFR were largely unaffected (Fig. EV3F). While this does not rule out a contribution of WASH to intracellular trafficking, this data strongly indicated a major contribution of WASH to virus endocytosis. In corroboration, HPV16 infection was also reduced upon silencing in HaCaT keratinocytes (Fig. EV3A,B), completely abrogated in CRISPR/Cas9-derived mouse fibroblast WASH knock-out (KO) cells (NIH-3T3, Figs. 4F,G and EV3D), and strongly reduced in human osteosarcoma WASH KO cells (U2OS) (Fig. EV3C,D). This not only confirmed the importance of WASH for cargo uptake, but also underlined the functional existence of

WASH-mediated endocytosis in cells derived from different tissues and species.

On endosomes, WASH exerts its function in retrograde transport in conjunction with the retromer complex (Cullen and Steinberg, 2018; Gomez and Billadeau, 2009; Harbour et al, 2012; Jia et al, 2012; Tu and Seaman, 2021; Wang et al, 2018). As previously observed, siRNA-mediated depletion of the retromer proteins vacuolar protein sorting (Vps) 26 and Vps29 interfered with HPV16 infection (Figs. 4C and EV4A) (Lipovsky et al, 2013; Popa et al, 2015; Zhang et al, 2018). However, since Vps29 is essential for HPV16 retrograde transport to the Golgi but expendable for virus uptake (Lipovsky et al, 2013; Popa et al, 2015; Xie et al, 2020), retromer itself is unlikely to facilitate endocytosis. Recent evidence suggests that the SNX-BAR adapter consisting of SNX1/2 and SNX5/6 can act independently of the retromer on endosomes (Kvainickas et al, 2017; Simonetti et al, 2017; Simonetti et al, 2019; Yong et al, 2020), potentially aided by endosomal protein termed receptor-mediated endocytosis 8 (RME-8), which coordinates WASH activity with the

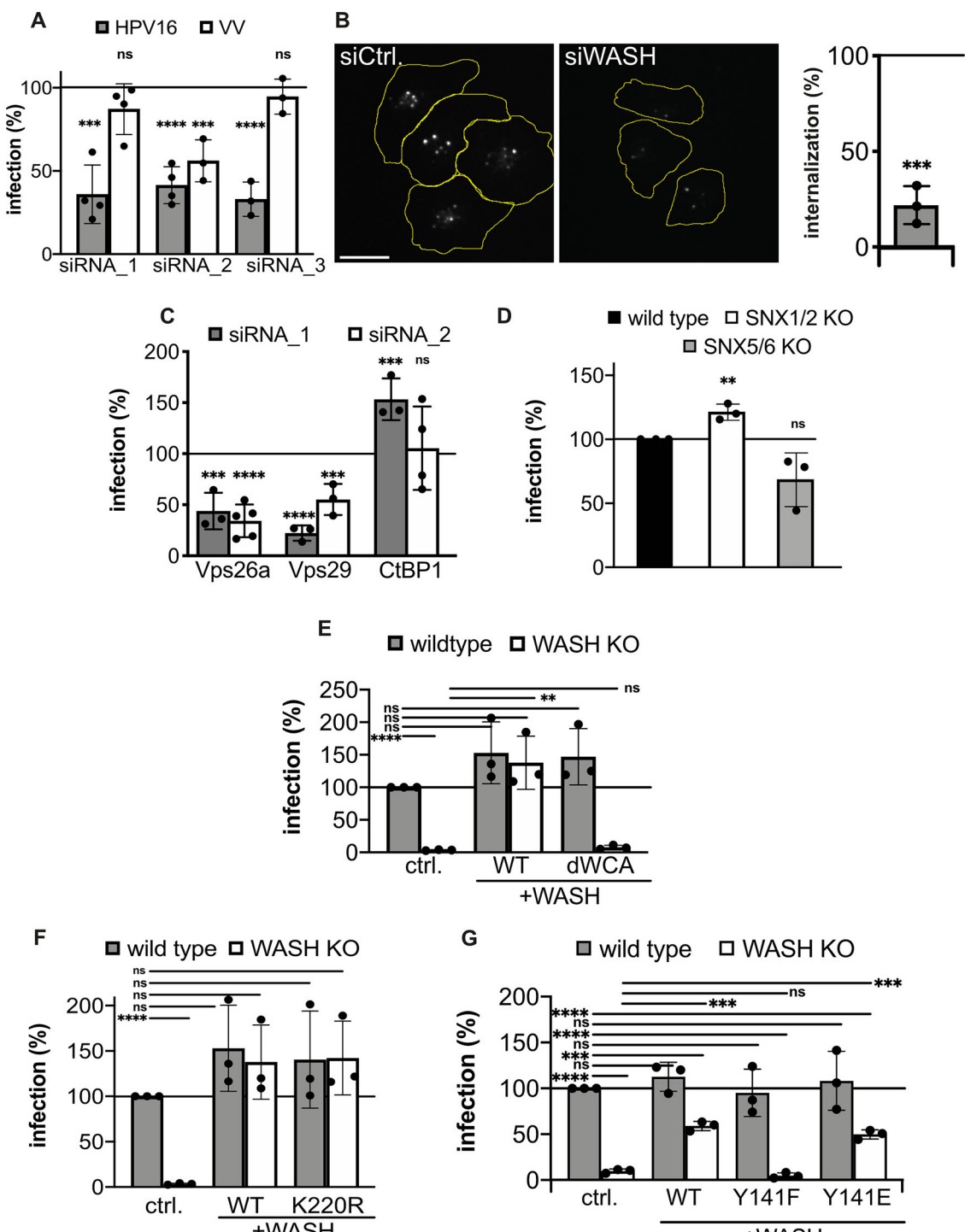

membrane bending retromer adapter SNX-BAR dimer (Freeman et al, 2014). Hence, the SNX-BAR dimer and RME-8 were depleted to assess whether they play an additional role during endocytosis. As control and to differentiate the mechanism studied here from macropinocytosis, depletion of CtBP1, a prominent BAR domain containing regulator of macropinocytosis (Liberali et al, 2008; Valente et al, 2013), did not impair HPV16 infection (Figs. 4C and EV4A). RNAi of RME-8 showed

that it was dispensable for HPV16 infection (Fig. EV4B). To assess the role of the SNX-BAR dimer, we employed CRISPR/Cas9 knockout cells of SNX1/SNX2 and SNX5/6 (Simonetti et al, 2017). Since SNX1/SNX2 and SNX5/6 knockout cells were unimpaired for HPV16 infection (Fig. 4D), they were also dispensable. These findings implied that WASH exerted its function during endocytosis in a way distinct from how it typically acts at endosomes.

◄

**Figure 4. WASH is controlled by Y141 for HPV16 uptake.**

(A) HeLa Kyoto cells were depleted of WASH and infected with HPV16-GFP or VV-GFP. Infection was determined 48 h p.i. by automated microscopy and flow cytometry and normalized to cells transfected with a control (ctrl.). (B) After depletion of WASH, cells were infected with HPV16-pHrodo and imaged live by spinning disk microscopy at 6 h p.i. Shown are average intensity projections of the HPV16-pHrodo signal with cell outlines (yellow), scale bar is 25 μm. The intensity of the virus signal per cell was normalized to ctrl. (C) HeLa Kyoto cells were depleted of Vps26a, Vps29, or CtBP1 and infected with HPV16-GFP. Infection was scored 48 h p.i. by automated microscopy and normalized to ctrl. (D) HeLa wild-type, SNX1/2 KO or SNX5/6 KO cells were infected with HPV16-GFP. Infection was scored 48 h p.i. by flow cytometry. (E–G) EGFP, WASH-YFP or the WASH mutants dWCA (E), K220R (F) or Y141F/E (G) were expressed in NIH-3T3 wild-type and WASH KO cells. Infection with HPV16-RFP was scored at 48 h p.i. by flow cytometry and normalized to wild-type cells expressing EGFP (ctrl.). Note that (E, F) are from the same experiment (same ctrl.) and that WT WASH did not fully restore HPV16 infection in (G) due to shorter expression times compared to (E, F). Data information: In (A–G), data are represented as mean ± SD. Dots in bar graphs indicate the replicates. For all quantifications, statistical significance was assessed by Student's $t$ test (*$P \le 0.05$, **$P \le 0.01$, ***$P \le 0.001$, ****$P \le 0.0001$, ns = not significant). For (A), data of $n = 3$ or $n = 4$ biological replicates. Statistical significance values are HPV16: siRNA_1: $P = 0.0003$, siRNA_2: $P \le 0.0001$, siRNA_3: $P \le 0.0001$; VV: siRNA_1: $P = 0.1435$, siRNA_2: $P = 0.0008$, siRNA_3: $P = 0.344$. In (B), data of $n = 3$ biological replicates with a significance value of $P = 0.0002$. In (C), data of $n = 3$ biological replicates with significance values of Vps26a – siRNA_1: $P = 0.0003$, siRNA_2: $P \le 0.0001$; Vps29 - siRNA_1: $P \le 0.0001$, siRNA_2: $P = 0.0004$; CtBP1 - siRNA_1: $P = 0.0008$, siRNA_2: $P = 0.7697$. In (D), data of $n = 3$ biological replicates. SNX1/2 KO: $P = 0.0042$, SNX5/6 KO: $P = 0.0599$. In (E), data of $n = 3$ biological replicates. Significance values in comparison to wild-type ctrl.: WASH KO ctrl.: $P \le 0.0001$; WASH WT: $P = 0.1251$; WASH KO: $P = 0.1852$; dWCA WT: $P = 0.1319$; dWCA KO: $P \le 0.0001$. Significance values in comparison to WASH KO ctrl.: WASH KO + WASH WT: $P = 0.0047$; WASH KO + WASH dWCA: $P = 0.088$. In (F), data of $n = 3$ biological replicates. Significance values in comparison to wild-type ctrl.: WASH KO-ctrl.: $P \le 0.0001$; wild-type ctrl. + WASH WT: $P = 0.1251$; $P \le 0.0001$; WASH KO ctrl. + WASH WT: $P = 0.1852$; wild-type ctrl. + WASH K220R: $P = 0.2598$; WASH KO ctrl. + WASH K220R: $P = 0.1459$. In (G), data of $n = 3$ biological replicates. Significance values in comparison to wild-type ctrl.: KO-ctrl.: $P \le 0.0001$; wild-type ctrl. + WASH WT: $P = 0.242$; WASH KO: $P = 0.0001$; wild-type ctrl. + WASH Y141F: $P = 0.7564$; WASH KO ctrl. + WASH Y141F: $P \le 0.0001$; wild-type ctrl. + WASH Y141E: $P = 0.6811$; WASH KO ctrl. + WASH Y141E: $P \le 0.0001$. Significance values in comparison to WASH KO ctrl.: WASH KO ctrl. + WASH WT: $P = 0.0001$; WASH KO ctrl. + WASH Y141E: $P = 0.0002$; WASH KO ctrl. + WASH Y141F: $P = 0.0833$.

## WASH acts during endocytosis largely independently of the regulatory complex at endosomes

Since our findings indicated a distinct WASH function for endocytosis compared to endosomal sorting, we next wondered whether WASH acted in concert with members of the WASH complex (SHRC), namely family with sequence similarity 21 (FAM21), strumpellin, strumpellin- and WASH-interacting protein (SWIP), and coiled-coil domain containing 53 (CCDC53). RNAi of the individual members of the complex showed that while SWIP, strumpellin, and CCDC53 were dispensable, FAM21 partially contributed to HPV16 infection (Fig. EV3H). Hence, WASH acted during HPV16 infection not through the SHRC, distinguishing it from its function on endosomes.

## WASH facilitates endocytosis by regulating actin polymerization

This raised the question of whether the function of WASH during HPV16 uptake was facilitated by its function as an actin nucleation-promoting factor. Ectopic expression of wild-type WASH rescued HPV16 infection in WASH knockout cells (Fig. 4E–G), whereas expression of WASH lacking the WASP-homology 2, central, and acidic (WCA) domain crucial for Arp2/3 complex activation did not (Fig. 4E). This indicated that WASH was an essential actin regulator of endocytosis.

Next, we investigated whether WASH was activated by ubiquitylation of residue K220, as previously described for its endosomal function (Hao et al, 2013). However, a WASH K220R mutant incapable of ubiquitylation rescued loss of WASH to the same extent than wild-type WASH (Fig. 4F) indicating that ubiquitylation of WASH was dispensable for WASH activation during endocytosis. In contrast, loss of function mutation of a phosphorylation site at Y141 previously implicated in WASH activity in natural killer (NK) cells (Huang et al, 2016) failed to rescue loss of WASH, whereas the gain of function mutation Y141E rescued similar to wild-type WASH (Fig. 4G). Again, this data distinguished WASH regulation during endocytosis from its typical function on endosomes.

## WASH acts during late stages of endocytosis

We then addressed the role of WASH during HPV16 endocytosis more specifically. Analysis of virus-containing pits in ultra-thin section transmission EM revealed that endocytic pits were morphologically unaltered in WASH KO compared to wild-type cells, i.e., they were fully formed and partially constricted at the neck (Fig. 5A). Thus, early stages of endocytic vesicle formation, such as induction and membrane invagination, were independent of WASH. Strikingly, the average number of virus-containing plasma membrane invaginations increased in the absence of WASH (Fig. 5A). Hence, endocytosis was stalled at a late stage such as scission in WASH KO cells. Why, however, did loss of WASH fail to replicate the phenotype of globally perturbed actin polymerization by cytoD, where virus-filled tubules formed (compare Fig. 1)? We reasoned that since cytoD interferes not only with WASH-mediated but also with cortical actin polymerization in general, cortical actin may restrict membrane feeding into endocytic pits. In this case, cytoD treatment in WASH knockout cells should replicate the earlier result. And in fact, virus-filled tubules arose also in the case of WASH knockout cells when treated with cytoD (Fig. 5B). Taken together, the roles of actin polymerization in vesicle scission, of WASH as the single NPF required to promote actin polymerization, and the loss of WASH stalling endocytosis at a late stage of vesicle formation strongly indicated that WASH regulated endocytic vesicle scission but not endocytic pit formation.

To directly stimulate vesicle scission, WASH would have to be present at endocytic sites. Recruitment of WASH to specialized TEMs marked by CD151 (Scheffer et al, 2013) was probed by a proximity ligation assay (PLA) (Söderberg et al, 2006). A small fraction of virus particles co-localized with the PLA signal of WASH and CD151 (Fig. EV5A). Hence, WASH may indeed act at sites of cargo uptake. Only a limited association of HPV16 with WASH/CD151 structures occurred within coincidence detection limits in fixed samples. This was expected, since virus uptake occurs asynchronously over many hours, so that only few viruses interact with specialized TEMs at any given time (Becker et al, 2018; Schelhaas et al, 2012). However, to more conclusively demonstrate

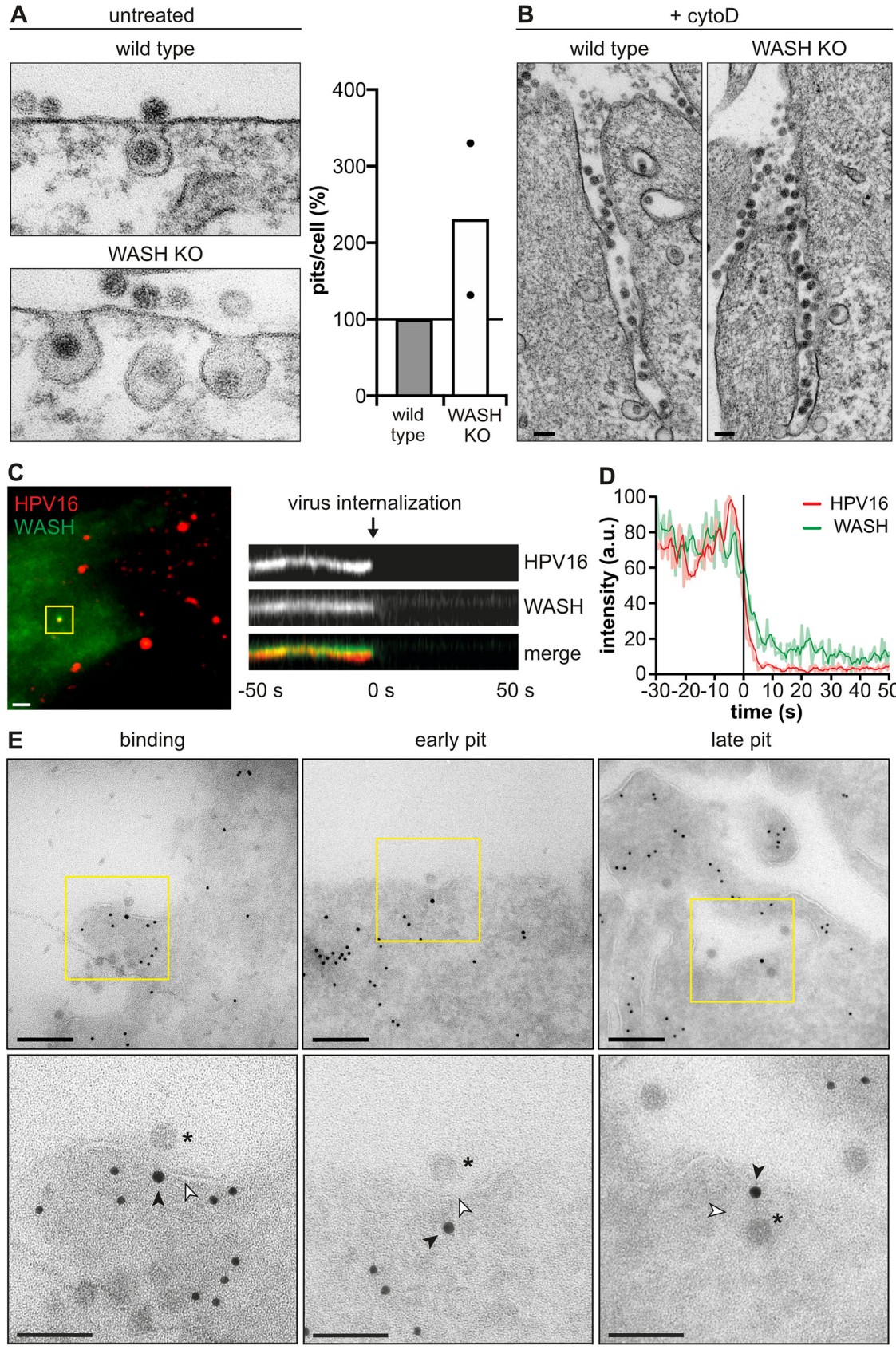

**Figure 5. WASH associates early, but acts late in endocytosis.**

(A) NIH-3T3 wild-type and WASH KO cells infected with HPV16-GFP were subjected to ultra-thin section transmission EM at 6 h p.i. The number of virus-filled plasma membrane invaginations was determined for 43 cells per cell line for $n = 2$ biological replicates. Total pit numbers were normalized to wild-type cells and are depicted as mean. Dots indicate the replicates. Scale bars are 100 nm. Note that while pits appear slightly bigger in the case of WASH KO cells, on average the variation of pit sizes between NIH3T3 and WASH KO cells is insignificant. (B) Infection of NIH-3T3 wild-type and WASH KO cells was carried out in the presence of 5 µg/ml cytoD. At 6 h p.i., cells were processed for ultra-thin section transmission EM. (C) HeLa ATCC cells transfected with EGFP-WASH were infected with HPV16-AF647. Cells were imaged by live-cell TIRF-M at 1 h p.i. Movies were acquired with 0.5 Hz frame rate for 5 min. HPV16 entry events were identified manually after background subtraction and filtering. Shown is a kymograph of the virus entry event highlighted by the yellow box, and the corresponding EGFP-WASH signal. Scale bar is 5 µm. (D) Intensity profiles of HPV16 and WASH (light red/green) as well as moving averages (intense red/green) of the virus entry event shown in (C) depicted relative to the half-time of virus loss ($t = 0$). (E) HeLa ATCC cells were transfected with EGFP-WASH and infected with HPV16. At 6 h p.i., cells were subjected to immunogold labeling of GFP (WASH, 15 nm gold) and actin (10 nm gold) on ultra-thin cryosections analyzed by transmission EM. Asterisks indicate HPV16 particles, black and white arrowheads indicate WASH staining and the membrane, respectively. Data information: (A, B) Scale bars are 100 nm. (C) Scale bar is 5 µM. (E) Scale bars are 200 nm and 100 nm for overviews and insets, respectively.

recruitment of WASH to endocytic sites and to gain information on recruitment dynamics, WASH association in relation to cargo uptake was analyzed by live-cell TIRF-M. WASH was detected together with HPV16 for more than 50 seconds prior to uptake and co-internalized with the cargo (Fig. 5C,D; Movie EV3). To pinpoint the stage during which WASH is recruited to endocytic sites, immunogold labeling and EM were performed. Indicative of specific labeling, WASH was found on endosomes (Fig. EV5B). Moreover, it was observed close to virus particles associated with flat plasma membrane regions (binding), slightly curved membranes (early pit), and with fully formed endocytic pits (late pit) (Fig. 5E). In fact, quantification of the immunogold EM revealed that WASH signals at the plasma membrane were similarly specific as WASH labeling on endosomes (Fig. EV5C). In conclusion, WASH was recruited to the plasma membrane already at a very initial stage of pit formation, and remained during all stages of pit maturation (Fig. 5E). Thus, WASH was recruited already early during vesicle formation, although it likely exerted its NPF function only during scission similar to its function on endosomes (Derivery et al, 2009; Gomez and Billadeau, 2009).

## Discussion

Here, we elucidate the mechanistic details of the endocytic process by which HPV16 is internalized into cells. Originally distinguished from other modes of endocytosis predominantly in negative terms, e.g., clathrin-, caveolin-, dynamin-, cholesterol-independent, and morphologically distinct from macropinocytosis, this work now provides evidence for the existence of a WASH-mediated scission mechanism during clathrin-independent endocytosis of HPV16. Cargo is internalized by inward budding of the plasma membrane in distinguishable steps. Thus, vesicles are most likely formed by de novo assembly of endocytic machinery in a modular manner, a mode comparable to CME (Fig. 6). Despite similar requirements for the sodium/protin antiporter and actin polymerization, the inward-budding pit formation renders WASH-mediated endocytosis of HPV16 clearly distinct from macropinocytosis (Fig. 6). While recent work claims a role of the WAVE complex prior to endocytic uptake by actin-mediated transport processes at the cell surface during HPV entry (Fernandez et al, 2025; Schelhaas et al, 2008a), and while the WASH complex may aid HPV16 infection also by intracellular sorting at the retromer, our results implicated WASH in the stimulation of branched actin polymerization for scission for HPV16 endocytosis. Importantly, the role of WASH in virus uptake was clearly distinguishable from its role in retrograde

trafficking due to its independence of retromer-associated sorting complexes, ubiquitylation at K220, and members of the WASH complex (SHRC). Hence, our work describes an unexpected direct function of WASH in endocytosis. Since WASH has not been attributed to any other endocytic pathway, it defines the molecular identity of this endocytic process.

The endocytic landscape includes a variety of processes, where endocytic vesicle formation is achieved by different machineries for different purposes. These typically diverge in morphology, key cargos, and molecular features (Doherty and McMahon, 2009). Often, the pinocytic mechanisms are subdivided into CME and clathrin-independent endocytosis, the latter of which applies to HPV uptake. Clathrin-independent endocytosis includes mostly mechanisms requiring cholesterol and glycosphingolipids, and the completion of vesicle formation by a scission event often depends on dynamin. Since HPV endocytosis is independent of cholesterol, glycosphingolipids, and dynamin (Schelhaas et al, 2012; Schowalter et al, 2011; Spoden et al, 2008; Spoden et al, 2013), it likely uses separate machinery.

Generally, scission may occur by one of several hypothesized mechanisms involving e.g., mechanoenzymatic scission by dynamin, membrane-lipid-reorganizing proteins or complexes (actin cytoskeleton and CtBP/BARS), or the appearance of a lipid diffusion barrier combined with a pulling force induced by molecular motors (friction-driven fission) (summarized in Renard et al, 2018). In mechanoenzymatic scission, e.g., during CME, dynamin polymerizes to form a collar that compresses the vesicle neck by an extensively discussed mechanism leading to membrane fission (Hinshaw and Schmid, 1995; Morlot and Roux, 2013; Sweitzer and Hinshaw, 1998; Takei et al, 1995), while HPV uptake occurs independently of dynamin (Schelhaas et al, 2012; Spoden et al, 2008) indicating that mechanoenzymatic scission by dynamin is dispensable for endocytic vesicle formation. HPV uptake involves actin polymerization, which coincided with vesicle scission in a timely fashion that resembled dynamin recruitment during CME (Merrifield, 2004; Merrifield et al, 2002). Hence, actin polymerization likely facilitates vesicle scission by a force-driven mechanism rather than as an anchoring structure for other scission factors. Alternative roles for actin polymerization in scission appear less likely, including membrane-lipid-reorganization followed by line tension-dependent scission as described for Shiga toxin (Romer et al, 2007) or as lipid diffusion barrier (friction-driven fission) as proposed by Simunovic and colleagues (2017): The first typically involves cholesterol-rich membranes and the second molecular motors such as dynein or myosin II, whereas HPV uptake requires neither (Schelhaas et al, 2008b; Schelhaas et al, 2012; Spoden et al, 2008; Spoden et al, 2013). Moreover, a hallmark of line tension-dependent

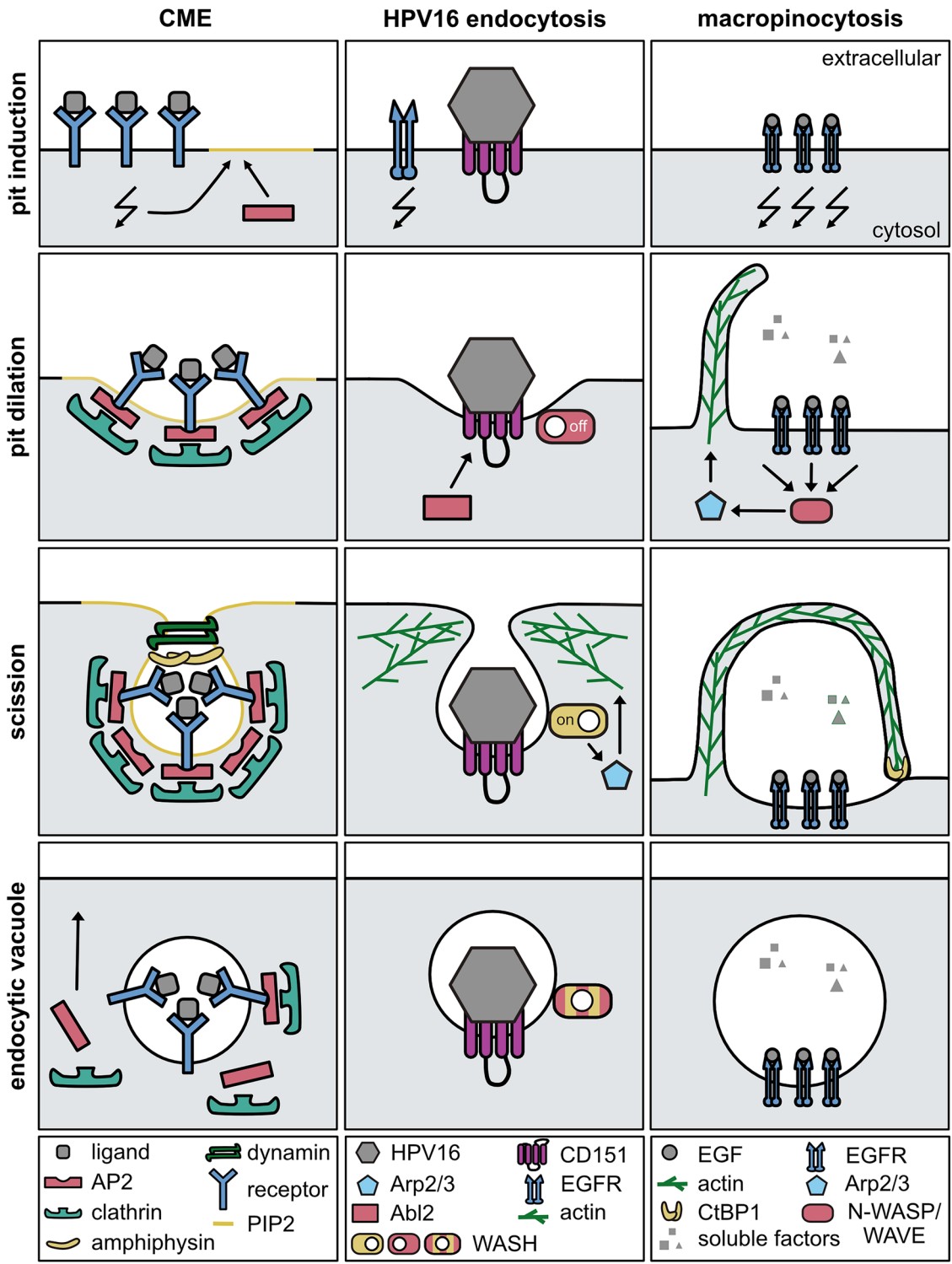

**Figure 6. Model.**

Schematic model of the mechanistic regulation of endocytic vesicle formation during HPV16 endocytosis in comparison to CME and macropinocytosis. Additional regulators involved in the latter mechanisms were omitted for clarity.

scission and friction-driven fission is the formation of tubular invaginations, which were absent upon WASH knockout. While this does not exclude that such mechanisms contribute to the fidelity of scission, particularly in light of cortical actin polymerization, it renders a force-driven scission mechanism through WASH-promoted actin polymerization more likely. How actin polymerization creates the force for vesicle scission remains elusive. However, in analogy to observations from CME or endocytosis in *Xenopus* oocytes, actin polymerization towards the vesicle neck may serve to constrict and propel the vesicle away from the plasma membrane (Bement et al, 2003; Collins et al, 2011; Sokac et al, 2003).

It remains unclear how WASH would be recruited to the plasma membrane. WASH recruitment to endosomes is regulated by incorporation into the WASH regulatory complex, similar to SNX1 or SNX2 recruitment as a heterodimeric complex with SNX5 or 6 (Derivery et al, 2009; Jia et al, 2010; Wassmer et al, 2007; Wassmer et al, 2009). However, since WASH can act independently of the SHRC for HPV endocytosis, and since SNX-BARs are dispensable in this context, WASH recruitment is likely to be different. For WASH, recruitment may be facilitated by FAM21 through direct phospholipid binding, as in vitro studies revealed FAM21 binding to a variety of phospholipids (Derivery et al, 2009; Jia et al, 2010).

Dissimilar to WASH activation at endosomes (Hao et al, 2013), ubiquitination of WASH was not essential for HPV endocytosis, whereas an ill-defined gain of function of WASH mediated by phosphorylation of Y141 that has been previously described in NK cells (Huang et al, 2016) may be important in this context. In NK cells, phosphorylation of WASH is mediated by lymphocyte-specific tyrosine kinase (Lck), which is not expressed in fibroblasts or epithelial cells. Instead, perhaps the related Abl2 kinase, previously implicated in HPV endocytosis, may function instead to activate WASH by phosphorylation of Y141 (Bannach et al, 2020).

How HPV uptake is induced by ligand–receptor interactions is an important question. While it is generally assumed that HPV–receptor interactions trigger the formation of endocytic pits due to the requirement of signal transduction for uptake (Bannach et al, 2020; Schelhaas et al, 2012; Survilaze et al, 2012), it remains to be conclusively substantiated. In fact, the existence of several virus particles in tubular endocytic structures points to the existence of plasma membrane sorting of viruses to preferred entry sites, which and may involve tetraspanin-enriched microdomains (Fast et al, 2018; Mikuličić et al, 2019; Scheffer et al, 2013; Spoden et al, 2008). As both scenarios are in fact not mutually exclusive, this question remains open for now.

From a more general perspective, it is tempting to speculate that WASH-mediated endocytosis occurs more prominently during specific cell states, as WASH is infrequently observed at the plasma membrane under normal cell culture conditions. Evolutionarily, it is probable that HPV16 exploits a cellular process that is easily inducible or already active upon infection of target cells. This situation exists upon epidermal wounding, where basal keratinocytes of the mucosal epidermis as primary targets become accessible to the virus (Aksoy et al, 2017; Doorbar, 2005; Roberts et al, 2007). Wounding and other cellular responses induce epithelial to mesenchymal transition and cell migration, which are accompanied by the remodeling of cell-matrix adhesion complexes, such as focal adhesions and hemidesmosomes (HDs) (Borradori and Sonnenberg, 1999; Ezratty et al, 2005; Jones et al, 1998; Walko et al, 2015; Webb et al, 2002). To date, relatively little is known about the dynamics of HD remodeling during cell migration, but HD-containing plasma membrane domains are rapidly endocytosed upon detachment from the underlying extracellular matrix (ECM) (Owaribe et al, 1990). Since HDs contain integrin α6 and CD151, which are part of the specialized TEMs that mediate HPV16 endocytosis (Mikuličić et al, 2019; Scheffer et al, 2013; Walko et al, 2015), the endocytic machinery facilitating HPV uptake may also promote HD uptake during wound healing to aid cell migration and thereby wound closure. In line with this notion, CD151 TEMs co-internalize with the virus and are remodeled upon growth factor signaling (Mikuličić et al, 2019; Scheffer et al, 2013). Interestingly, a CME sorting motif in the cytosolic domain of CD151 is not required for its function during HPV endocytosis, supporting its specialized role for retrieval of functionalized TEMs (Liu et al, 2007; Scheffer et al, 2013). Another essential process during wound healing is matrix remodeling. Thus, it is noteworthy that WASH was previously implicated in apical endocytosis of extracellular material in the *Drosophila* airway epithelium during embryogenesis (Tsarouhas et al, 2019). Hence, WASH-mediated endocytosis may also contribute to matrix remodeling as a specialized mechanism for internalization of extracellular material.

While future research will have to address the cellular role of HPV endocytosis in more detail, its role in pathogen invasion may be of considerable interest as well. For instance, WASH is recruited to the plasma membrane during *Salmonella* infection, presumably serving as one of several entry pathways (Hänisch et al, 2010). Moreover, WASH regulatory complex protein FAM21 co-localizes with a subset of VV particles in plasma membrane lipid rafts (Huang et al, 2008). Thus, HPV endocytosis may constitute an important entry route for pathogens. Accordingly, viruses such as IAV and LCMV exploit pathways with endocytic vacuole morphology and mechanistic requirements similar yet not identical to HPV16 endocytosis (de Vries et al, 2011; Quirin et al, 2008; Rojek et al, 2008; Sieczkarski and Whittaker, 2005).

# Methods

## Cell lines

HeLa ATCC, HeLa Kyoto, HeLa SNX1/2 KO, HeLa SNX5/6 KO (all female origin), and HaCaT cells (male origin) were cultured in

**Reagents and tools table**

| Reagent/resource | Reference or source | Identifier or catalog number |
|---|---|---|
| **Experimental models** | | |
| **cell lines** | | |
| Hamster: BHK-21-Helsinki | Kind gift from A. Helenius (ETH Zurich, Switzerland) (Johannsdottir et al, 2009) | N/A |
| African green monkey: BSC-40 | Kind gift from J. Mercer (UCL London, UK) (Brockman and Nathans, 1974) | N/A |

| Reagent/resource | Reference or source | Identifier or catalog number |
|---|---|---|
| Human: HaCaT | Boukamp et al, 1988 | N/A |
| Human: HeLa ATCC | ATCC (Manassas, USA) | N/A |
| Human: HeLa Kyoto | Kind gift from L. Pelkmans (ETH Zurich, Switzerland) (Landry et al, 2013) | N/A |
| Human: HeLa SNX1/2 KO | Kind gift from P. Cullen (University of Bristol, UK) (Simonetti et al, 2017) | N/A |
| Human: HeLa SNX5/6 KO | Kind gift from P. Cullen (University of Bristol, UK) (Simonetti et al, 2017) | N/A |
| Mouse: NIH 3T3 | ATCC (Manassas, USA) | N/A |
| Mouse: NIH 3T3 WASH KOs | This study | N/A |
| Human: U2OS | German Collection of Microorganisms and Cell Cultures (DSMZ, Braunschweig, Germany) | N/A |
| Human: U2OS WASH KOs | this study | N/A |
| Human: 293TT | Kind gift from J. Schiller (NIH, Bethesda, USA) (Buck et al, 2005) | N/A |
| **Bacteria and Viruses** | | |
| *E. coli* DH5a | Kind gift from A. Helenius (ETH Zurich, Switzerland) | N/A |
| HPV16-GFP pseudoviruses (PsVs) | (Buck and Thompson, 2007) | N/A |
| HPV16-RFP PsVs | Johnson et al, 2009 | N/A |
| SFV (prototype strain) | Kind gift from A. Helenius (ETH Zurich, Switzerland) (Marsh and Helenius, 1980) | N/A |
| VSV-GFP (Indiana) | Kind gift from A. Helenius (ETH Zurich, Switzerland) (Johannsdottir et al, 2009) | N/A |
| VV-GFP (Western Reserve) | Kind gift from J. Mercer (UCL London, UK) (Mercer and Helenius, 2008) | N/A |
| **Recombinant DNA** | | |
| Plasmid: pSpCas9(BB)-2A-GFP (pX458) | Addgene | 48138 |
| Plasmid: pEGFP C1_CD151 | Kind gift from L. Florin (Johannes Gutenberg University Mainz, Germany) (Liu et al, 2007) | N/A |
| Plasmid: pcDNA3 HA_CD151 | This study | N/A |
| Plasmid: pmRFP C3_Clc | Kind gift from A. Helenius (ETH Zurich, Switzerland) | N/A |
| Plasmid: pEGFP-Dyn2 | Kind gift from M. A. McNiven (Mayo Clinic, Rochester, USA) (Cao et al, 1998) | N/A |
| Plasmid: pClneoEGFP | Kind gift from C. Buck (NIH, Bethesda, USA) (Buck and Thompson, 2007) | N/A |
| Plasmid: Lifeact_EGFP-N1 | Kind gift from R. Wedlich-Söldner (University of Münster, Germany) (Riedl et al, 2008) | N/A |
| Plasmid: pEGFP C1 | Kind gift from A. Helenius (ETH Zurich, Switzerland) | N/A |
| Plasmid: pRwB | Kind gift from C. Buck (NIH, Bethesda, USA) (Johnson et al, 2009) | N/A |
| Plasmid: pEGFP C1_Scar-W-GFP | Machesky and Insall, 1998 | N/A |
| Plasmid: pEGFP-C1_Scar-WA-GFP | Machesky and Insall, 1998 | N/A |
| Plasmid: pcDNA5_EGFP-WASH | Derivery et al, 2009 | N/A |
| Plasmid: pCMS3H1p YFP WASH wild-type | Kind gift from D. Billadeau (Mayo Clinic, Rochester, USA) (Gomez and Billadeau, 2009) | N/A |
| Plasmid: pCMS3H1p YFP WASH dWCA | Kind gift from D. Billadeau (Mayo Clinic, Rochester, USA) (Gomez and Billadeau, 2009) | N/A |
| Plasmid: pCMS3H1p YFP WASH K220R | Kind gift from P. R. Potts (UT Southwestern Medical Center, Dallas, USA) (Hao et al, 2013) | N/A |
| Plasmid: pCMS3H1p YFP WASH Y141F/E | This study | N/A |
| Plasmid: p16SheLL | Kind gift from C. Buck (NIH, Bethesda, USA) (Buck et al, 2005) | N/A |
| **Antibodies** | | |
| Mouse anti-actin (4F7) | Braunschweig Integrated Centre for Systems Biology https://www.tu-braunschweig.de/brics (Schroeder et al, 2009) | bsbs300470 |
| Rabbit polyclonal anti-Arp3 | Steffen et al, 2006 | N/A |
| Rabbit polyclonal anti-CCDC53 | Merck | ABT69 |
| Mouse monoclonal (11G5a) anti-CD151 | Bio Rad | MCA1856GA |

| Reagent/resource | Reference or source | Identifier or catalog number |
| --- | --- | --- |
| Mouse monoclonal (3) anti-CtBP1 | Becton Dickinson | 612042 |
| Mouse monoclonal (R-1) anti-EGFR | Santa Cruz | sc101 |
| Rabbit polyclonal anti-ERK2 (C-14) | Santa Cruz Biotechnology | sc-154 |
| Rabbit polyclonal anti-FAM21C | Merck | ABT79 |
| Rabbit polyclonal anti-GFP | Abcam | ab290 |
| Mouse monoclonal anti-HA (16B12) | Covance | MMS-101R |
| Goat polyclonal anti-JMY (L-16) | Santa Cruz Biotechnology | sc-10027 |
| Rabbit monoclonal (11H13L5) anti DNAJC13 | Invitrogen | 702773 |
| Primary rabbit antiserum "Lady Di" against SFV E1/2 | kind gift from A. Helenius (ETH Zurich, Switzerland) (Singh and Helenius, 1992) | N/A |
| Rabbit polyclonal anti-strumpellin | Santa Cruz | sc-87442 |
| Rabbit polyclonal anti-SWIP | Merck | ABT70 |
| Mouse monoclonal (B-5-1-2) anti-α-tubulin | Sigma-Aldrich | T5168 |
| Mouse monoclonal (DM1A) anti-α-tubulin | Sigma-Aldrich | T6199 |
| Rabbit polyclonal anti-Vps26a | Abcam | Ab137447 |
| Rabbit polyclonal anti-Vps29 | Abcam | Ab98929 |
| Rabbit polyclonal anti-WASH | Kind gift from A. Gautreau (Institut Polytechnique De Paris, Paris, France) (Derivery et al, 2009) | N/A |
| Rabbit polyclonal anti-WASH | Atlas Antibodies | HPA002689 |
| Rabbit polyclonal anti-WAVE1 | Sigma-Aldrich | W2142 |
| Rabbit polyclonal anti-WAVE2 (H-110) | Santa Cruz Biotechnology | sc-33548 |
| Rabbit polyclonal anti-WHAMM (K-13) | Santa Cruz Biotechnology | sc-136951 |
| Goat anti-mouse AF488 IgG | Thermo Fisher Scientific | A-11029 |
| Goat anti-mouse AF647 IgG | Thermo Fisher Scientific | A-21235 |
| Goat anti-rabbit AF488 IgG | Thermo Fisher Scientific | A-11034 |
| Rabbit anti-mouse IgG + IgM | Dianova | 315-005-048 |
| Sheep anti-mouse HRP-linked IgG | GE Healthcare | NA931 |
| Donkey anti-rabbit HRP-linked IgG | GE Healthcare | NA934 |
| Duolink In Situ PLA Probe anti-mouse PLUS | Sigma-Aldrich | DUO92001 |
| Duolink In Situ PLA Probe anti-rabbit MINUS | Sigma-Aldrich | DUO92005 |
| Protein A gold 10 nm | CMC, UMC Utrecht | N/A |
| Protein A gold 15 nm | CMC, UMC Utrecht | N/A |
| **Oligonucleotides and other sequence-based reagents** | | |
| Single guide RNA (sgRNA) WASH exon 2 (murine, top strand) | GCGACGAGAGGAGGCAATCC | N/A |
| sgRNA WASH exon 2 (murine, bottom strand) | GGATTGCCTCCTCTCGTCGC | N/A |
| sgRNA WASH exon 4 (human, top strand) | TCTTCACGGGCGCCCAGGAC | N/A |
| sgRNA WASH exon 4 (human, bottom strand) | GTCCTGGGCGCCCGTGAAGA | N/A |

| Reagent/resource | Reference or source | Identifier or catalog number |
|---|---|---|
| sgRNA WASH exon 5 (human, top strand) | GTGTGCGTGAGCACCAAGCC | N/A |
| sgRNA WASH exon 5 (human, bottom strand) | GGCTTGGTGCTCACGCACAC | N/A |
| Murine WASH sequencing primer (forward) | ATAGGCAGAGGGGTGAGTGT | N/A |
| Murine WASH sequencing primer (reverse) | ACACTGGGCATTAGTTGGGT | N/A |
| M13r primer for TOPO vector | CAGGAAACAGCTATGAC | N/A |
| WASH Y141E mutagenesis primer (forward) | GAA GCT GAA AGA GTT TCC TGT GTG TGT GAG CAC CAA GC | N/A |
| WASH Y141E mutagenesis primer (reverse) | GCT TGG TGC TCA CAC ACA CAG GAA ACT CTT TCA GCT TC | N/A |
| WASH Y141F mutagenesis primer (forward) | GAAGCTGAAGTTCTTTCCTGTGTGTGTGAGCACCAAGC | N/A |
| WASH Y141F mutagenesis primer (reverse) | GCTTGGTGCTCACACACACAGGAAAGAACTTCAGCTTC | N/A |

| Reagent/resource | Reference or source | Working concentration (nM) | Identifier or catalog number |
|---|---|---|---|
| **siRNAs** | All from Qiagen | | |
| AllStars death | | 10 | SI04381048 |
| AllStars negative (siCtrl.) | | 10/20/50 | SI03650318 |
| Arp3 siRNA_1 | | 10 | Hs_ACTR3_5 |
| Arp3 siRNA_2 | | 10 | Hs_ACTR3_8 |
| CCDC53 siRNA_1 | | 10 | Hs_CCDC53_3 |
| CCDC53 siRNA_2 | | 10 | Hs_CCDC53_4 |
| CtBP1 siRNA_1 | | 10 | Hs_CTBP1_5 |
| CtBP1 siRNA_2 | | 10 | Hs_CTBP1_6 |
| DIAPH2 siRNA_1 | | 10 | Hs_DIAPH2_1 |
| DIAPH2 siRNA_2 | | 10 | Hs_DIAPH2_4 |
| DIAPH2 siRNA_3 | | 10 | Hs_DIAPH2_6 |
| FAM21 siRNA_1 | | 10 | Hs_FAM21_11 |
| FAM21 siRNA_2 | | 10 | Hs_FAM21_12 |
| FNL2 siRNA_1 | | 10 | Hs_FMNL2_6 |
| FNL2 siRNA_2 | | 10 | Hs_FMNL2_7 |
| FNL2 siRNA_3 | | 10 | Hs_FMNL2_8 |
| FNL3 siRNA_1 | | 10 | Hs_FMNL3_1 |
| FNL3 siRNA_2 | | 10 | Hs_FMNL3_5 |
| FNL3 siRNA_3 | | 10 | Hs_FMNL3_6 |
| Formin 1 siRNA_1 | | 10 | Hs_FMN1_5 |
| Formin 1 siRNA_2 | | 10 | Hs_FMN1_6 |
| Formin 1 siRNA_3 | | 10 | Hs_FMN1_7 |
| Formin 2 siRNA_1 | | 10 | Hs_FMN2_12 |
| Formin 2 siRNA_2 | | 10 | Hs_FMN2_7 |
| Formin 2 siRNA_3 | | 10 | Hs_FMN2_9 |
| GFP siRNA | | 10 | SI04380467 |
| JMY siRNA_1 | | 50 | Hs_JMY_1 |

| Reagent/resource | Reference or source | Identifier or catalog number |
|---|---|---|
| JMY siRNA_2 | 50 | Hs_JMY_5 |
| N-WASP siRNA_1 | 10 | Hs_WASL_1 |
| N-WASP siRNA_2 | 10 | Hs_WASL_6 |
| RME8 siRNA_1 | 50 | Hs_DNAJC13_1 |
| RME8 siRNA_2 | 50 | Hs_DNAJC13_4 |
| RME8 siRNA_3 | 50 | Hs_DNAJC13_5 |
| RME8 siRNA_4 | 50 | Hs_DNAJC13_6 |
| strumpellin siRNA_1 | 10 | Hs_KIAA0196_7 |
| strumpellin siRNA_2 | 10 | Hs_KIAA0196_8 |
| SWIP siRNA_1 | 10 | Hs_KIAA1033_5 |
| SWIP siRNA_2 | 10 | Hs_KIAA1033_7 |
| WASH siRNA_1 | 20 | Hs_WASH1_4 |
| WASH siRNA_2 | 20 | Hs_WASH1_8 |
| WASH siRNA_3 | 20 | Hs_WASH1_7 |
| WAVE1 siRNA_1 | 20 | Hs_WASF1_3 |
| WAVE1 siRNA_2 | 50 | Hs_WASF1_4 |
| WAVE2 siRNA_1 | 10 | Hs_WASF2_5 |
| WAVE2 siRNA_2 | 10 | Hs_WASF2_6 |
| WHAMM siRNA_1 | 50 | Hs_WHDC1_1 |
| WHAMM siRNA_2 | 50 | Hs_WHDC1_2 |
| Vps26 siRNA_1 | 10 | Hs_VPS26A_1 |
| Vps26 siRNA_2 | 10 | Hs_VPS26A_2 |
| Vps29 siRNA_1 | 10 | Hs_VPS29_6 |
| Vps29 siRNA_2 | 10 | Hs_VPS29_2 |
| **Reagent/resource** | **Reference or source** | **Identifier or catalog number** |
| **Chemicals, enzymes, and other reagents** | | |
| AGAR 100 Resin kit (Epoxy resin) | Agar Scientific | R1031 |
| Alexa Fluor 488 succinimidyl ester | Thermo Fisher Scientific | A20000 |
| Alexa Fluor 568 succinimidyl ester | Thermo Fisher Scientific | A20003 |
| Alexa Fluor 594 succinimidyl ester | Thermo Fisher Scientific | A20004 |
| Alexa Fluor 647 succinimidyl ester | Thermo Fisher Scientific | A20006 |
| Bafilomycin A1 | Applichem | A7823 |
| Brij-58 | Sigma-Aldrich | P5884 |
| Cytochalasin D | Sigma-Aldrich | C8273 |
| Dynasore | Merck | 324410 |
| EIPA | Sigma-Aldrich | A3085 |
| Glutaraldehyde, EM grade | Polysciences | 01909 |
| Heparin | Sigma-Aldrich | H4784 |
| Lipofectamine 2000 | Invitrogen | 11668019 |
| Lipofectamine RNAi max | Invitrogen | 13778075 |
| Osmium tetroxide | Electron Microscopy Sciences | 19100 |
| Paraformaldehyde, EM grade | Polysciences | 00380 |

| Reagent/resource | Reference or source | Identifier or catalog number |
|---|---|---|
| Phalloidin-Atto488 | Sigma-Aldrich | 49409 |
| pHrodo Red succinimidyl ester | Thermo Fisher Scientific | P36600 |
| QuikChange II XL Site-Directed Mutagenesis kit | Agilent Technologies | 200516 |
| RedDot2 | VWR | 40061-1 |
| ROCK inhibitor | Becton Dickinson | Y-27632 |
| SMIFH2 | Sigma-Aldrich | S4826 |
| Trypan Blue solution (0,4%) | Thermo Fisher Scientific | 15250061 |
| uranyl acetate | Polysciences | 21447 |
| X-tremeGENE™ 9 | Merck | XTG9-RO |
| Duolink In Situ Detection Reagents Red | Sigma-Aldrich | DUO92008 |
| **Software and algorithms** | | |
| Adobe Illustrator | Adobe Inc. | Version 16.0.4 |
| Adobe Photoshop | Adobe Inc. | C4, extended version 11.0 |
| Affinity Designer | Serif (Europe) Ltd | Version 1.7.3 |
| ApE – A plasmid Editor | M. Wayne Davis (https://jorgensen.biology.utah.edu/wayned/ape/) | Version 1.11 |
| BD CellQuest Pro | Becton Dickinson | Version 5.2 |
| CellProfiler | Kamentsky et al, 2011 | Version 2.1.1 |
| CellSens Dimension | Olympus | Version 2.3 |
| FACSDiva | Becton Dickinson | Version 6.1 |
| Fiji (ImageJ) | Schindelin et al, 2012 | Versions 2.0.0/1.50 g and 2.1.0/1.53c |
| FlowJo | Becton Dickinson | Versions 8.8.6 and 10.6.1 |
| guavaSoft | Merck | Version 3.1.1 |
| iTEMFEI | FEI/Olympus | Version 5.2 |
| MATLAB | MathWorks | Version R2015a, 8.5.0.197613 |
| MATLAB InfectionCounter Program | Snijder et al, 2012 | Version blue, release B3 |
| Metamorph (Spinning disk) | Molecular Devices | Version 7.7.2 |
| Metamorph (IX71 TIRF) | Molecular Devices | Version 7.7.1 |
| Microsoft Excel | Microsoft Corporation | Versions 15.41.0 and 16.40 |
| Prism | GraphPad Software, Inc. | Version 6.0f |
| SnapGene Viewer | SnapGene | Version 2.6.2 |
| Tecnai software | FEI/Thermo Fisher Scientific | Version 3.1.3 |
| Volocity software | PerkinElmer | Version 6.3 |

high-glucose Dulbecco's modified Eagle medium (DMEM, D5796 Sigma-Aldrich) supplemented with 10% fetal bovine serum (FBS). 293TT cells (female origin) were grown in DMEM with 10% FBS and 400 μg/μl hygromycin B. U2OS wild-type and WASH KO cells (female origin) as well as murine NIH 3T3 wild-type and WASH KO cells (male origin) were cultured in DMEM supplemented with 10% FBS and 1% non-essential amino acids (NEAA). Primate, non-human BSC-40 cells (sex unspecified) were grown in DMEM supplemented with 10% FBS, 5% NEAA, and 5% sodium pyruvate. Hamster BHK-21-Helsinki cells (male origin) were cultured in

Glasgow's minimum essential medium (GMEM) supplemented with 10% FBS. All cells were cultivated in a humidified atmosphere at 37 °C and 5% $CO_2$ and regularly authenticated and tested for mycoplasma contamination.

## Bacteria strains

Chemocompetent *E. coli* DH5a used for plasmid preparation was grown in lysogeny broth (LB) medium supplemented with antibiotics at 37 °C and 210 rpm.

## Virus strains

VV-GFP (Western Reserve) containing a fluorescent version of the core protein A5 was propagated and titrated on BSC-40 cells in standard growth medium and purified as described previously (Mercer and Helenius, 2008). SFV (prototype strain) was propagated and titrated on BHK-21-Helsinki cells as previously described (Marsh et al, 1984). Infection was carried out in GMEM supplemented with 10% FBS and 10% Tryptose Broth. VSV-GFP (Indiana) expressing an additional transgene encoding GFP was propagated on BHK-21-Helsinki cells grown in GMEM supplemented with 10 mM HEPES (pH 6.5). At 1 h p.i., GMEM supplemented with 30 mM HEPES (pH 7.3) and 10% Tryptose Phosphate Broth and 1% FBS was added, as previously described (Johannsdottir et al, 2009). The virus was titrated on BHK-21-Helsinki cells grown in RPMI supplemented with 30 mM HEPES (pH 6.5). HPV16 PsVs were produced as described in "Methods".

## Cloning and plasmid purification

### Cloning of HA-CD151 construct
Lentiviral expression constructs of HA-tagged CD151 were a kind gift from M. Hemler (Dana Farber Cancer Institute and Harvard Medical School, Boston, USA) (Hwang et al, 2019). For subcloning, HA-CD151 was cut from the lentiviral vector using *Eco*RI and *Xba*I for 2 h at 37 °C. A pcDNA3 expression vector was cut using the same enzymes and conditions. For purification, both samples were run on a 2% acrylamide gel. The DNA was visualized with ethidium bromide, and bands representing the cut pcDNA3 backbone as well as HA-CD151 (insert) were isolated from the gel. Gel extraction was performed with a NucleoSpin Gel and PCR Clean-up kit (Macherey-Nagel). DNA concentrations were determined with the help of a 1.5% agarose gel by comparison to the marker bands (Gene Ruler 1 kb DNA ladder, Thermo Scientific). The insert and backbone were ligated by incubation with the T4 ligase overnight at 16 °C.

### Plasmid purification
Chemocompetent *E. coli* DH5a were incubated with 5 μl ligation product for 10 min on ice. Heat shock was performed by 60–90 s incubation at 42 °C. Immediately afterwards, bacteria were incubated on ice for 5 min before LB medium was added. Bacteria were grown at 37 °C and 350 rpm for 30–60 min and plated on LB agar plates with antibiotics using inoculation loops. Plates were incubated overnight at 37 °C and inspected for colony growth the next day. For HA-CD151 cloning, several overnight cultures with LB supplemented with antibiotics were inoculated with one colony each. Plasmids were purified using the NucleoSpin Plasmid kit (Macherey-Nagel) and sent for sequencing by Eurofins Genomics (Luxembourg). Sequence analysis was performed with ApE. For use in experiments, plasmids purified using the NucleoBond Xtra Maxi kit (Macherey-Nagel).

### Site-directed mutagenesis
WASH Y141 mutants were generated following the protocol of the QuikChange II XL Site-Directed Mutagenesis kit (Agilent Technologies). Primers were designed based on the manufacturer's guidelines and are listed in the key resources table. Following the PCR and DpnI digest, chemocompetent *E. coli* DH5a were

transformed, and plasmids were purified as described above. Successful mutagenesis was controlled by PCR.

### sgRNA design and cloning
sgRNAs were provided by a CRISPR design tool (CRIPR.mit.edu or CCTop). Specifically, the murine gene was disrupted by sgRNAs targeting exon 2, and the human gene by simultaneously targeting exons 4 and 5 (key resources table). Top and bottom strands of sgRNAs were annealed using T4 ligase for 30 min at 37 °C and cloned into pSpCas9(BB)-2A-GFP (pX458) by digestion with *Bbs*I and ligation with T4 ligase (6 cycles: 37 °C for 5 min, 21 °C for 5 min). Residual linearized DNA was removed by treatment with PlasmidSafe exonuclease at 37 °C for 30 min. Chemically competent *E.coli* DH5a were transformed with the ligation product as described above.

## Generation and characterization of WASH KO cell lines with CRISPR/Cas9

NIH-3T3 and U2OS cells were plated in six-well plates and maintained in DMEM (4.5 g/L glucose, Invitrogen, Germany) supplemented with 10% (v/v) FBS (Sigma, Germany), 1 mM sodium pyruvate, 1× non-essential amino acids, and 2 mM L-glutamine at 37 °C in a humidified 7.5% CO2-atmosphere overnight. Cells were genome edited using the CRISPR/Cas9 technology (Ran et al, 2013) to generate WASH KO cell lines. Selected sgRNAs (key resources table) were cloned into pX458 allowing simultaneous expression of sgRNA, Cas9 and selection via EGFP expression as described above. The resulting constructs were transfected into NIH-3T3 or U2OS cells, respectively. Plasmids (1 μg) and X-tremeGene™ 9 (3 μl) were diluted in 100 μl optiMEM, incubated for 30 min at room temperature and added to cells for 16–24 h. Transfection efficiency was monitored using an EVOS® FL Cell Imaging System (Thermo Fisher, Germany). Cells were grown to confluence, and subsequently single, GFP-positive cells were sorted into 96-well plates by flow cytometry using a FACSAria II instrument (BD Biosciences) and FACSDiva software. Sorted cells were maintained in growth medium supplemented with penicillin (50 Units/ml)/streptomycin (50 mg/ml) (Thermo Fisher Scientific) and containing 30% conditioned medium and 10 μM ROCK inhibitor (BD Biosciences). After ~10 days, clones were picked from single wells and expanded. Derived cell clones were screened for the absence of WASH expression by western blotting, and NIH-3T3 clones lacking detectable amounts of WASH were subjected to genomic sequencing as described (Kage et al, 2017). Cells from confluent 6 cm dishes were trypsinized, pelleted, and lysed by overnight incubation in lysis buffer (100 mM Tris pH 8.5, 5 mM EDTA, 0.2% SDS, 200 mM NaCl, 20 mg/ml proteinase K) at 55 °C. Nucleic acid extraction was performed by ethanol precipitation. The addition of 700 μl 100% ice-cold ethanol was followed by centrifugation at 16,000 × g at 4 °C for 30 min. The pellet was washed with 400 μl 70% ice-cold ethanol, and samples were dried at 45 °C for 20 min. DNA was dissolved in 100 μl deionized water at 4 °C overnight and served as template in PCR using GoTaq G2 flexi DNA polymerase. The selected primer pair (key resources table) revealed a PCR product of 330 base pairs. PCR products were examined on 2% agarose gels and appropriate samples purified with a NucleoSpin Gel and PCR clean-up kit (Macherey-Nagel). DNA fragments were cloned into a zero-blunt TOPO vector (Zero Blunt

TOPO Cloning Kit for Sequencing, Invitrogen) for 5 min at room temperature. After transformation as above, single bacterial colonies were inoculated overnight and plasmid DNA purified using NucleoSpin Plasmid kit (Macherey-Nagel). Sequencing of isolated plasmid DNA was carried out by MWG-Biotech using a standard M13r sequencing primer (key resources table). Clones were examined for frameshift mutations and mono- or biallelic deletions/insertions using SnapGene Viewer software. Mutations or deletions generating stop codons shortly downstream of the target site were defined as "null" alleles. Cell populations exclusively harboring such alleles out of more than 50 sequencing reactions were selected for further analyses. For WASH targeted clones of murine NIH-3T3 cells, three clones were identified that did not display a wild-type allele in more than ten sequencing reactions. All clones showed similar effects on HPV16 infection, thus only one clone is shown in this study. For WASH targeted clones of human U2OS cells no clear sequencing result was obtained probably due to pseudogenes being targeted by sequencing primers.

## Western blotting

For analysis of protein amounts after siRNA treatment or CRISPR/Cas9-mediated KO, lysates were prepared in 2× sample buffer (4% SDS, 20% glycerol, 0.01% bromophenol blue, 100 mM Tris HCl (pH 6.8), 200 mM DTT). Samples were denatured for 5 min at 95 °C and loaded on polyacrylamide gels. For stacking gels, 5% polyacrylamide was used, and separating cells were 6 or 8% for JMY and WHAMM, respectively. All other proteins were separated on 10% gels. Electrophoresis was performed in Laemmli running buffer (0.1% SDS, 25 mM Tris, 192 mM glycine). Proteins were transferred from gels to nitrocellulose membranes for 50 min at 400 mA in pre-cooled transfer buffer (192 mM glycine, 25 mM Tris, 10% methanol). After transfer, membranes were blocked in 5% milk powder (MP) in Tris-buffered saline (TBS) supplemented with 0.2% Tween 20 (TBS-TMP) or in 0.2-3% bovine serum albumin (BSA) for at least 30 min. Primary antibodies were diluted in TBS-TMP or BSA and membranes were incubated for 2 h at room temperature or overnight at 4 °C. Three washes with TBS supplemented with Tween 20 (TBS-T) were followed by incubation with anti-mouse or anti-rabbit secondary antibodies conjugated to HRP diluted in TBS-TMP or BSA. Membranes were washed twice with TBS-T and once with TBS before the signal from HRP-conjugated antibodies was revealed using enhanced chemiluminescence (ECL) or ECL prime and photographic films or a chemiluminescence imager (ChemoStar Touch, Intas).

## Virus production

### Production of HPV16 PsVs

PsV production was performed as previously described (Buck et al, 2005). A total of $1.8 \times 10^7$ 293TT cells were seeded in 145 mm cell culture dishes. The next day, cells were co-transfected with p16sheLL and the reporter plasmid pClneoEGFP (GFP) or pRwB (RFP). Both plasmids (30 μg each) as well as Lipofectamine 2000 (132.5 μl) were diluted in optiMEM and incubated for 5 min at room temperature. The DNA dilution was added to the Lipofectamine 2000 dilution and samples were incubated for 20 min at room temperature before the transfection mix was added to fresh growth medium in the dishes. At 48 h post transfection,

cells were harvested and pelleted. For cell lysis and virus maturation, the pellet was incubated with 0.35% Brij 58, 0.2% Plasmid Safe DNase, and 0.2% benzonase for 24 h at 37 °C on an overhead rotator. PsVs were purified using a linear 25–39% OptiPrep density gradient. A PsV fraction at around 30% OptiPrep was collected with a needle and analyzed for virus content and purity by Coomassie staining of SDS-PAGE gels.

### Labeling of HPV16 PsVs

HPV16 PsVs were incubated with Alexa Fluor 488, 568, 594, or 647 succinimidyl ester in virion buffer (635 mM NaCl, 0.9 mM $CaCl_2$, 0.5 mM $MgCl_2$, 2.1 mM KCl in PBS, pH 7.6) using a 1:8 molar ratio of L1 to the dye for 1 h on an overhead rotator (Samperio Ventayol and Schelhaas, 2015; Schelhaas et al, 2008b). Free dye was removed by ultracentrifugation using a 15–25–39% OptiPrep step gradient. The labeled virus between the 25% and 39% OptiPrep fraction was collected with a needle. The PsV concentration was determined by SDS-PAGE and subsequent Coomassie staining. The labeled virus was characterized by binding to glass coverslips and HeLa ATCC cells (Samperio Ventayol and Schelhaas, 2015). Labeling of PsVs with pHrodo was achieved following the same protocol. Virus characterization was performed in citric acid buffer (pH 4.4) (Samperio Ventayol and Schelhaas, 2015).

## Infection experiments

### Infection of KO cells

NIH-3T3 ($4 \times 10^5$ cells/well) and U2OS (both $5 \times 10^4$ cells/well) wild-type and WASH KO cells were seeded in 12-well plates. The next day, the growth medium was replaced with 300 μl fresh growth medium, and HPV16-GFP was added to result in 20% infection of wild-type cells (about 30 ng and 15 ng, respectively). The virus was bound on a shaker at 37 °C. At 2 h p.i., the inoculum was replaced by fresh growth medium and infection was continued. At 48 h p.i., cells were trypsinized and fixed in 4% PFA for 15 min at room temperature. Cells were resuspended in FACS buffer (250 mM EDTA, 2% FBS, 0.02% $NaN_3$ in PBS) and analyzed for infection (percentage of GFP-positive cells) by flow cytometry (FACSCalibur, Becton Dickinson). Gating of infected cells was done with the help of uninfected controls. The percentage of infected cells was normalized to the respective wild-type cells using Microsoft Excel.

### Inhibitor studies

HeLa ATCC were seeded in 12-well plates ($5 \times 10^4$ cells/well) about 16 h prior to experimentation. For HPV16 infection experiments, small compound inhibitors and solvent controls were diluted in growth medium, while they were diluted in infection medium (RPMI supplemented with 30 mM HEPES, pH 6.5) for VSV infection. Cells were pre-treated with inhibitors or solvent controls for 30 min at indicated concentrations and infected with HPV16-GFP as described above. The inoculum was replaced at 2 h p.i. and infection was continued in the presence of the inhibitor. At 12 h p.i., inhibitors were exchanged for 10 mM $NH_4Cl$/10 mM HEPES in growth medium to reduce cytotoxicity (Schelhaas et al, 2012). Cells were fixed and processed for flow cytometry analysis as described above. For infection with VSV-GFP, the virus was added to the infection medium +/− inhibitor to result in 20% infection in solvent-treated controls. VSV-GFP was bound for 2 h on a shaker

at 37 °C until the inoculum was replaced with 1 ml growth medium. Cells were trypsinized and fixed at 6 h p.i. as described for HPV16.

### Infection studies after siRNA-mediated depletion

For siRNA-mediated knockdown, $2 \times 103$ or $2 \times 10^4$ HeLa Kyoto and HaCaT cells were reverse-transfected in 96-well optical bottom plates or 12-well plates, respectively. Transfection was performed using 0.2 μl (96-well) or 0.5 μl (12-well) Lipofectamine RNAi max per well diluted in optiMEM and siRNAs were diluted in optiMEM to reach the working concentration (see Reagents and Tools Table). The following procedure and incubation times were as for Lipofectamine 2000. Besides the siRNA against the cellular proteins of interest, the AllStars-negative siRNA (ctrl.) was included as a non-targeting control, whereas the AllStars death siRNA was used to test for successful transfection. Moreover, an siRNA targeting GFP was included to suppress the expression of the HPV16-GFP reporter plasmid as a measure for maximal reduction of infection. For RNAi against WASH, cells were transfected twice in 48 h intervals. Cells were infected with HPV16-GFP at 48 h post transfection to result in 20% infection in ctrl. negative transfected controls (about 7 ng/well). In 12-well plates, infection was performed and analyzed by flow cytometry as described above (using about 12 ng HPV16-GFP/well). Absolute infection values were normalized to ctrl. transfected controls. In 96-well plates, the virus was added without prior medium exchange to reduce cell loss. At 48 h p.i., cells were fixed in 4% PFA in PBS and nuclei were stained with RedDot2 for 30 min after permeabilization with 0.1% Triton in PBS. Infection was analyzed by automated microscopy on a Zeiss Axiovert Z.1 microscope equipped with a Yokogawa CSU22 spinning disk module (Visitron Systems). Images were acquired using a 20x objective, a CoolSnap HQ camera (Photometrics) and MetaMorph Software. Cell number and infection were determined using a MATLAB-based infection scoring procedure (Engel et al, 2011). The program detects nuclei and infection signal individually, based on their limiting intensity edges. The edges were filled to objects, which were classified by size. Binary masks of nuclei and infection signal were created and cells were classified as infected if equal or greater than 5% of their nuclei overlapped with infection signal above a certain threshold. In this study, signal twice above the background in the uninfected sample was considered infected. An infection index was obtained for each image and averaged per well (Snijder et al, 2012).

Infection with VV-GFP was carried out in 96-well plates following the same protocol as for HPV16 with the exception that $3 \times 10^3$ cells were transfected. Virus amounts leading to 20% infection in ctrl.-treated controls were used (about 0.5 plaque-forming units/well). Cells were fixed at 6 h p.i. and analyzed by automated microscopy, as described above.

For siRNA experiments with SFV, $5 \times 10^4$ HeLa Kyoto cells were reverse-transfected in 12-well plates. Infection was performed by addition of the virus to infection medium (RPMI supplemented with 10% FBS, 10 mM HEPES (pH 7.3)). The virus was bound on a shaker at 37 °C (about 0.5 plaque-forming units/well). At 2 h p.i., the inoculum was replaced by growth medium. Cells were trypsinized and fixed at 6 h p.i. Since SFV did not carry a fluorescent reporter plasmid, samples were immunostained for SFV E1/E2 after fixation at 6 h p.i. Cells were permeabilized with FACS perm (250 mM EDTA, 2% FBS, 0.02% NaN₃, 0.05% Saponin in PBS) for 30 min at room temperature and subsequently incubated

with the Lady Di antiserum (Singh and Helenius, 1992) diluted in FACS perm for 2 h at room temperature. Samples were washed thrice with FACS perm and incubated with an anti-rabbit AF488 secondary antibody in FACS perm for 1 h at room temperature. Washing with FACS perm was followed by infection scoring with FACS analysis (FACSCalibur, Becton Dickinson) as described for HPV16. Infection values were normalized to crtl. using Microsoft Excel.

### Infection studies in transiently transfected cells

NIH-3T3 wild-type and WASH KO cells were seeded in 12-well plates ($4 \times 10^4$ cells/well). One day later, cells were transfected with plasmids (1 μg) encoding EGFP-WASH or EGFP, YFP-WASH, YFP-WASH dWCA, YFP-WASH K220R, YFP-WASH Y141F or YPF-WASH Y141E using Lipofectamine 2000 (0.5 μl/well) diluted in optiMEM. Incubation times were the same as for virus preparation. At about 16 h post transfection, cells were infected with HPV16-RFP (about 15 ng/well) as described above to result in 20% infection in control cells transfected with GFP. Cells were trypsinized at 48 h p.i., fixed in 4% PFA in PBS, and analyzed by flow cytometry (Guava easyCyte, Merck). Final analysis was performed with FlowJo. Transfected cells were gated with the help of untransfected controls. Then, the GFP-positive population was gated for infection using transfected, but uninfected controls. The percentage of transfected and infected cells (GFP and RFP positive) was normalized to NIH-3T3 wild-type cells transfected with the GFP control to obtain relative infection values using Microsoft Excel.

HeLa ATCC cells were transfected with Scar-W and -WA constructs as described above 16–24 h prior to infection. Cells were infected with HPV16-RFP and fixed 48 h p.i. using 4% PFA. Infection was scored using an Olympus IX70 inverted microscope equipped with an electron multiplier CCD camera (EDMCCD, C9100-02, Hamamatsu Photonics K.K.) and a monochromator for epifluorescence excitation. Images were thresholded manually and at least 100 cells were scored for transfection and infection using Fiji.

### HPV16 binding assay

For analysis of HPV16 binding by flow cytometry, $5 \times 10^4$ NIH-3T3 wild-type, WASH KO cells, or HeLa Kyoto cells transfected with siRNAs against WASH as described above were seeded per well of a 12-well plate. The next day, fluorescently labeled HPV16-AF488 (~1000 particles/cell) was bound to the cells for 2 h on a shaker at 37 °C. Cells treated with siRNAs were reseeded at 48 h post transfection ($5 \times 10^4$ cells/well). Virus binding was performed once cells were attached, typically about 6 h post seeding. As a non-binding control, HPV16-AF488 was pre-incubated with 1 mg/ml heparin for 1 h at room temperature prior to binding to cells (Cerqueira et al, 2013). Cells were trypsinized and fixed with 4% PFA. Virus binding was analyzed by measuring the mean fluorescence intensity (geometric mean) of cells in flow cytometry (FACSCalibur). The geometric mean of uninfected cells was subtracted from infected cells, and virus binding was normalized to control cells. A similar procedure was applied to measure virus binding to HeLa Kyoto cells depleted of SNX2 or SNX5. Binding was qualitatively assessed for NIH-3T3 wild-type and WASH KO cells. For this, HPV16-AF568 was bound as described above. At 2 h p.i., cells were fixed with 4% PFA and stained with 0.1 μg/ml

phalloidin-Atto488 diluted in PHEM buffer (60 mM PIPES, 10 mM EGTA, 2 mM MgCl$_2$, 25 mM HEPES, pH 6.9) supplemented with 0.01% Triton X-100 for 30 min. Cells were washed thrice with PBS and mounted on glass slides using AF1 mounting medium. Images were acquired with a Zeiss Axiovert Z.1 microscope equipped with a Yokogawa CSU22 spinning disk module (Visitron Systems) using a ×40 objective, a CoolSnap HQ camera (Photometrics), and MetaMorph Software. Z-stacks covering the cell volume were converted to maximum intensity projections using Fiji. Brightness and contrast were adjusted using uninfected samples.

### Endocytosis assay with HPV16-pHrodo

Cells were reverse-transfected with siRNAs as described above. At 48 h after the first (Arp3) or second (WASH) siRNA transfection, HPV16-pHrodo (~1000 particles/cell) was added to 350 µl growth medium and bound for 2 h on a shaker. The inoculum was replaced at 2 h p.i. and cells were imaged live at 6 h p.i. at 37 °C and 5% CO$_2$ in a humidified atmosphere using custom-made imaging chambers and DMEM high glucose without phenol red supplemented with 10% FBS, 1% L-glutamine and 1% penicillin/streptomycin. Images were acquired on a Zeiss Axiovert Z.1 microscope equipped with a Yokogawa CSU22 spinning disk module (Visitron Systems) using a ×40 objective, a CoolSnap HQ camera (Photometrics) and MetaMorph Software. Average intensity projections of confocal slices were generated using Fiji software. Intensity-based analysis was performed with CellProfiler (Becker et al, 2018). In brief, the virus signal was enhanced by application of a white top-hat filter. Virus spots were segmented by application of a Gaussian filter and maximum correlation thresholding. Virus intensity was then measured in enhanced and original images. Pivot tables (Microsoft Excel) were used to summarize the intensity of spots per condition. These values were normalized to the cell number, which was determined by manual counting from brightfield images. The virus intensity per cell was then normalized to ctrl. treated controls to obtain relative internalization values. Cell outlines were created manually for presentation purposes.

### Endocytosis assay by quenching of HPV16-AF594

About $5 \times 10^4$ HeLa Kyoto cells were reverse-transfected with ctrl. or WASH siRNAs as described above and seeded into 12-well plates. Cells were infected with HPV16 AF594 (about 1000 particles/cell) in the presence or absence of inhibitors. Cells were washed once with PBS, and lifted of the plates by adding cell dissociation buffer (Thermo Fisher Scientific) for 5 min. The cell dissociation buffer was replaced by PBS after gently sedimenting the cells by centrifugation, and cells were submitted to flow cytometry analysis after addition of trypan blue (0.4% (w/v), Thermo Fisher Scientific) in a 1:50 dilution. This immediately shifts the emission spectrum of viruses on the cell-surface and leads to a loss of detectable fluorescence, whereas the fluorescence of intracellular viruses remained unaltered (Schelhaas et al, 2012). Cells were analyzed by flow cytometry and the GeoMean of remaining cell-associated fluorescence was normalized to the ctrl., and depicted as relative internalization.

## Electron microscopy and CLEM

### CLEM

A circular mark used for localization of unroofed cells was generated in the center of coverslips (22 mm diameter) using a diamond knife. For ECM production, $2 \times 10^6$ HaCaT cells were seeded onto the coverslips placed in a six-well plate. At 48 h post seeding, ECM was obtained by detaching cells through incubation with 10 mM EDTA/EGTA for 45 min at 37 °C, subsequent clapping of the plate and washes with PBS (Culp et al, 2006). HPV16-AF568 was bound to the ECM in 1 ml growth medium/well for 2 h on a shaker at 37 °C. HaCaT cells were trypsinized and $12 \times 10^5$ cells/well were seeded onto the virus bound to ECM. At 1 h post seeding, 10 µg/ml cytoD or DMSO (solvent control) were added. A total of 6 h post seeding, cells were put on ice and washed thrice with cold stabilization buffer (70 mM KCl, 30 mM HEPES (pH 7.4 with KOH), 5 mM MgCl$_2$). For unroofing, the cells were kept on ice, and 1 ml cold 2% PFA in stabilization buffer was aspirated with a 1-ml pipette. The pipette was positioned above the marked area in the center of the coverslip, and PFA was harshly released onto the cells. The coverslip was then rapidly transferred to a new well containing cold 2% PFA in stabilization buffer to avoid sedimentation of cell debris on the unroofed membrane. Membrane sheets were fixed for 10 min at 4 °C. Samples were mounted in custom-made imaging chambers and imaged in PBS at a Nikon Ti Eclipse microscope equipped with a PerkinElmer UltraVIEW VoX spinning disk module. Images were acquired using a ×60 objective, an Orca Flash 4 camera (Hamamatsu) and Volocity software (PerkinElmer, version 6.3). Montages of $10 \times 10$ images and 10% overlap were acquired around the center of the marked area. The unroofed membranes were prepared for EM by fixation with 2% glutaraldehyde (GA) in PBS overnight at 4 °C. After two washes with water, samples were incubated with 0.1% tannic acid for 20 min at room temperature and subsequently washed with water. Contrasting was performed with 0.1% uranyl acetate (UAC) for 20 min at room temperature. After three washes with water, samples were dehydrated with a series of ethanol solutions (15%/30%/50%/70%/80%/90%/100%). Coverslips were incubated with each solution for 5 min, incubation with 100% ethanol was repeated thrice. Samples were dried using hexamethyldisilazane (HDMS). After 5 min incubation at room temperature, fresh HDMS was added and samples were incubated for a further 30 min at room temperature. Coverslips were dried and coated under vacuum using a Balzers BAF301 device (former Balzers AG, Liechtenstein). A first layer of platinum was applied at an angle of 11 °C while rotating. A second layer of carbon was applied at an angle of 90° while rotating. Coverslips were cut to fit on EM grids before 5% hydrofluoric acid were used to separate the metal replica from the glass. Replicas were extensively washed with water prior to transfer to glow-discharged, formvar-coated EM grids. Replicas were imaged with a phase contrast microscope for orientation. Intact membranes associated with virus particles were manually selected based on overlays of images from fluorescence and phase contrast microscopy and imaged at a transmission EM (Jeol JEM-1400, Jeol Ltd., Tokyo, Japan, camera: TemCam F416; TVIPS, Gauting, Germany). Membrane sheets were imaged using montages of $5 \times 5$ images and 15% overlap. Fluorescence and EM images were initially overlaid manually using Photoshop, then the Fiji plugin Landmark Correspondences (Saalfeld and Tomancák, 2008) was used for transformation of the fluorescence image according to the EM image using three manually identified landmarks. For analysis, HPV16 was identified manually based on the fluorescent signal and classification was done by visual evaluation of associated structures in transmission EM images. A total of 134 and 101 membrane-

associated virus particles in DMSO (7 membranes) and cytoD (5 membranes) treated cells were analyzed, respectively.

### Ultra-thin section EM

Samples for ultra-thin section EM were prepared and analyzed as described previously (Bannach et al, 2020). A total of $1–2 \times 10^5$ NIH 3T3 wild-type, WASH KO and HeLa Kyoto cells or were seeded in 3 cm dishes. Two days post seeding, cells were either pre-treated with inhibitors for 30 min or were left untreated. Cells were infected with 40 µg HPV16 PsVs in 1 ml growth medium. At 6 h p.i., cells were fixed in 2.5% GA in PBS (pH 7.2) for 10 min at room temperature. A second fixation was performed with cold 2.5% GA at 4 °C overnight. Cells were washed thrice with PBS, post-fixed with 1% $OsO_4$ in $ddH_2O$ for 1 h and washed twice with $ddH_2O$ at room temperature and at 4 °C for 20 min, respectively. Block contrasting was performed with 0.5% UAC in $ddH_2O$ at 4 °C overnight. Cells were washed thrice with $ddH_2O$ and dehydrated using an ascending graded alcohol series. Detaching and dehydrating with propylene oxide were followed by incubation with propylene oxide and epoxy resin (1:3, 1:1, 3:1) for 2 h each, before cells were incubated with pure epoxy resin for 3 h and embedded in BEEM capsules. The resin was allowed to polymerize at 60 °C for three days before 60 nm ultra-thin sections were cut and counter-stained with uranyl acetate and lead. Samples were imaged at a 80 kV on a Tecnai 12 electron microscope (FEI) using an Olympus Veleta 4k CCD camera or Ditabis imaging plates. Images were contrast-enhanced and cropped using Adobe Photoshop CS4. The total number of endocytic pits per cell was determined for 31 and 43 cells in wild-type and WASH KO cells, respectively, in two independent experiments. Only endocytic pits containing virus(es) were counted, since HPV16 pits are hardly distinguishable from uncoated pits from other endocytic pathways without further staining. Pit numbers were normalized to wild-type cells.

### Immunogold labeling

HeLa ATCC cells ($2–3 \times 10^5$ cells) were seeded in 6 cm dishes. The next day, cells were transfected with a plasmid encoding EGFP-WASH or EGFP-SNX2 (7 µg) using Lipofectamine 2000 (3.5 µl) according to the manufacturer's instructions or left untransfected. At 48 h post transfection, cells were infected with 80 µg HPV16 PsVs and incubated for 6 h at 37 °C until pre-fixation by the addition of 4% formaldehyde in 0.1 M phosphate buffer (pH 7.2) to the culture medium (1:1 ratio) for 5 min. Then, cells were fixed in 2% formaldehyde and 0.2% GA in 0.1 M phosphate buffer (pH 7.2). Samples were processed for transmission EM as previously described (Humbel and Stierhof, 2009). In brief, cells were quenched by incubation in 0.1% glycine in 0.1 M PB ($2 \times 30$ min), washed thrice in 0.1 M PB for 30 min, and scraped with 1% gelatin. After centrifugation, the gelatin was replaced with 12% gelatin, and cells were infused at 37 °C. Cells were cooled down on ice and the gelatin-cell pellet was cut into small blocks that were infused with 2.3 M sucrose overnight at 4 °C. The blocks were mounted on specimen carriers and frozen in liquid nitrogen. Ultra-thin cryosections were prepared according to Tokuyasu (Tokuyasu, 1980). In brief, ultra-thin cryosections of 50 nm thickness were prepared with an EM UC6/FC6 ultramicrotome (Leica Micro-systems). Sections were collected in a sucrose-methylcellulose mixture and stored on transmission EM grids at 4 °C until further processing. Methylcellulose was melted and sections were washed

five times with 20 mM glycine in PBS. Quenching was followed by blocking with 1% BSA for 3 min. Cells were then incubated with a GFP-antibody for 30 min, washed 6 times with 0.1% BSA in PBS, and incubated with protein A gold 15 nm for 20 min. Sections were rinsed ten times with PBS and re-fixed in 1% GA in PBS (pH 7.2). For double labeling, the sections were quenched, blocked, and immunostained with an actin antibody as described above. Then sections were incubated with a rabbit anti-mouse bridging antibody followed by six washes with 0.1% BSA in PBS, and incubated with protein A gold 10 nm (1:50). Sections were rinsed ten times with PBS and re-fixed in 1% GA in PBS, pH 7.2. Sections were rinsed ten times with $ddH_2O$ and contrasted with uranyl acetate for 6 min (pH 7). After one wash with $ddH_2O$, cells were embedded in an uranyl acetate-methylcellulose mixture (pH 4) for 10 min. After looping out with filter paper, sections were dried, and images were acquired as above.

For quantification, 50 cell profiles were randomly selected. Antibody signals were determined as the number of gold particles associated with specific cellular compartments. Gold particles counted in 50 profiles of HeLa cells treated with Lipofectamine 2000 (negative control) were used as control.

## Fluorescence microscopy

### CLC and virus internalization analysis by live-cell TIRF microscopy

HeLa ATCC cells were seeded on coverslips in 12-well plates ($5 \times 10^4$ cells/well) one day prior to transfection. Cells were transfected with plasmids encoding lifeact-EGFP, EGFP-WASH, EGFP-SNX2, mRFP-CLC or EGFP-Dyn2 as described above. For internalization analysis, fluorescently labeled virus (HPV16-AF594/AF647) was bound at 37 °C at about 18 h post transfection. At 1 h p.i., coverslips were mounted in custom-made imaging chambers. Cells were imaged at 37 °C and 5% $CO_2$ in humidified atmosphere in DMEM without phenol red supplemented with 10% FBS, 1% L-glutamine and 1% penicillin/streptomycin. Time-lapse movies of cells expressing lifeact-EGFP were acquired with a ×60 TIRF objective on an Olympus IX70 microscope equipped with a TIRF condenser and an electron multiplier CCD camera (EMCCD, C9100-02, Hamamatsu Photonics K.K.) using MetaMorph software (Molecular Devices) (Visitron Systems). All other time-lapse movies were acquired using a 100x TIRF objective at an Olympus IX83 microscope equipped with a four-line TIRF condenser and an EMCCD camera (iXon Ultra 888, Oxford Instruments) controlled by CellSens Dimensions software (Olympus). Movies were acquired with 0.5 Hz frame rate for 5 min. HPV16 entry events were identified manually and the intensity of fluorescent proteins at virus entry sites was analyzed after rolling ball background subtraction and filtering (mean intensity filter) with Fiji. Kymographs, intensity profiles along a manually drawn line through the virus/clathrin signal over time, were generated with Fiji after background subtraction and filtering. Intensity profiles measured with Fiji using a circular region of interest, were processed by min/max normalization and aligned by setting the half-time of virus internalization to 0 s. Profiles were plotted with Microsoft Excel. Moving averages of signals are shown as a trendline (period 4–20). The time points of recruitment onset, maximal signal or signal loss were manually determined relative to the half-time of internalization and box plots were generated with GraphPad Prism. Cells co-transfected with CLC and Dyn2 were analyzed the same way at

about 16 h post transfection. Movies were compressed to 20 fps and PNG.

### Proximity ligation assay

For ECM production on coverslips, $5 \times 10^5$ HaCaT cells were seeded per well of a 12-well plate and cultivated at 37 °C for 2 days. In addition, $7 \times 10^4$ HaCaT cells were seeded per well and transfected the next day with a plasmid encoding HA-CD151. The procedure was the same as for infection studies with transfected cells, but 0.4 μg DNA and 1 μl Lipofectamine 2000 per well were used. ECM on coverslips was obtained by detaching cells with 0.5 ml 10 mM EDTA/EGTA as described above. HPV16-AF488 (~1000 particles/cell) was bound to the ECM in 400 μl growth medium on a shaker at 37 °C. At 2 h post binding, HaCaT cells expressing HA-CD151 were trypsinized and transferred onto the virus bound to ECM. Cells were allowed to attach for 5 h and fixed in 2% PFA in PBS for 10 min at 4 °C. Cells were permeabilized with 0.2% Brij 58 in PBS for 10 min at room temperature prior to blocking in 1% BSA in PBS for 30 min at room temperature. Primary antibody staining against HA (1:10,000) and WASH (1:500) was carried out in a wet chamber overnight at 4 °C. Next, cells were incubated with anti-mouse PLUS and anti-rabbit MINUS Duolink PLA probes diluted 1:5 in 1% BSA in PBS for 1 h at 37 °C in a humidity chamber. Duolink In Situ Detection Reagents Red were used for further sample processing. For ligation of PLA probes, 1.25 U ligase per sample were added to the corresponding ligation buffer diluted in ddH$_2$O. The cells were incubated in a wet chamber for 30 min at 37 °C followed by two washes with wash buffer B (2 min, room temperature). Amplification was performed with 6.25 U polymerase and Duolink Amplification Red (1:5) diluted in ddH$_2$O in a wet chamber at 37 °C for 100 min. Cells were washed twice for 10 min with wash buffer B at room temperature, followed by a quick wash with 0.1× wash buffer B in ddH$_2$O. As a counterstain, HA (CD151) was detected with an anti-mouse AF647 antibody, which was diluted 1:2000 in 1% BSA in PBS. Samples were transferred to custom-made imaging chambers. Images were acquired with a 100x TIRF objective at an Olympus IX83 microscope equipped with a four-line TIRF condenser and an EMCCD camera (iXon Ultra 888, Andor Oxford Instruments) using CellSens Dimension software (Olympus).

### Flow cytometry analysis of receptor cell surface levels in U2OS WASH KO cells

U2OS WT and WASH KO cells were detached by incubation with 3 ml 50 mM EDTA for 10 min at 37 °C. After fixation with 2% PFA (4 °C, 10 min), cells were sedimented (5 min, $600 \times g$, 4 °C) and washed twice with PBS. Blocking was performed with 1% BSA for 1 h at room temperature. Subsequently, $7 \times 10^4$ cells were incubated with anti-EGFR (1:150) or CD151 (1:500) antibodies diluted in 1% BSA at 4 °C overnight. Three washes with PBS were followed by incubation with Alexa Fluor 488-conjugated secondary antibodies diluted in 1% BSA for 1 h at room temperature. Cells were washed thrice with PBS and resuspended in 300 μl flow cytometry buffer (250 mM EDTA, 2% FBS, 0.02% NaN3 in PBS). A Gallios Flow Cytometer (Beckman Coulter) was used to determine receptor cell surface levels as the geometric mean fluorescence intensity. Background values from controls incubated with secondary antibodies only were subtracted, and intensities were normalized to wild-type cells.

## Quantification and statistical analysis

Information on data representation (mean ± SD) and $n$ can be found in the figure legends. Statistical significance was determined using unpaired $t$ tests conducted with GraphPad Prism. $P$ values are indicated in the figure legends.

## Data availability

This study includes source data deposited at Biostudies in the following database record: https://www.ebi.ac.uk/biostudies/studies/S-BSST2069.

The source data of this paper are collected in the following database record: biostudies:S-SCDT-10_1038-S44319-025-00594-3.

## Peer review information

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

## Acknowledgements

We thank I Fels and D Kaiser (Institute of Cellular Virology, Münster, Germany) for technical support, and U Westerkamp and M Dominguez (Institute of Cellular Virology, Münster, Germany) for assistance with experiments. We are indebted to numerous individuals for sharing valuable reagents as indicated in the methods. We acknowledge the Infectious Diseases Imaging Platform at the University Hospital Heidelberg, K Richter from the Core Facility Unit Electron Microscopy at the DKFZ Heidelberg and S Hillmer from the Electron Microscopy Core Facility of the University Heidelberg for their technical support. This work was supported by funding of the German Research Foundation (DFG) to MS (grants SCHE 1552/6-2, INST211/10295-1, and INST211/817A09), and to SB (grant BO 4340/2-1). TEBS acknowledges support by the Helmholtz Society through HGF impulse fund W2/W3-066.

## Author contributions

**Pia Brinkert**: Conceptualization; Formal analysis; Investigation; Visualization; Methodology; Writing—original draft; Writing—review and editing. **Lena Krebs**: Conceptualization; Formal analysis; Investigation; Visualization; Methodology; Writing—original draft; Writing—review and editing. **Pilar Samperio Ventayol**: Formal analysis; Funding acquisition; Investigation; Methodology; Writing—review and editing. **Lilo Greune**: Formal analysis; Investigation; Methodology. **Carina Bannach**: Formal analysis; Investigation. **Cynthia Amakiri**: Formal analysis; Investigation; Methodology; Writing—review and editing. **Delia Bucher**: Formal analysis; Investigation; Methodology. **Jana Kollasser**: Resources; Formal analysis; Investigation; Methodology; Writing—review and editing. **Petra Dersch**: Resources; Formal analysis; Writing—review and editing. **Steeve Boulant**: Conceptualization; Resources; Formal analysis; Funding acquisition; Writing—review and editing. **Theresia E B Stradal**: Conceptualization; Resources; Formal analysis; Supervision; Funding acquisition; Investigation; Writing—original draft; Project administration; Writing—review and editing. **Mario Schelhaas**: Conceptualization; Resources; Formal analysis; Supervision; Funding acquisition; Investigation; Visualization; Methodology; Writing—original draft; Project administration; Writing—review and editing.

Source data underlying figure panels in this paper may have individual authorship assigned. Where available, figure panel/source data authorship is listed in the following database record: biostudies:S-SCDT-10_1038-S44319-025-00594-3.

## Funding

## Disclosure and competing interests statement

The authors declare no competing interests.

# Expanded View Figures

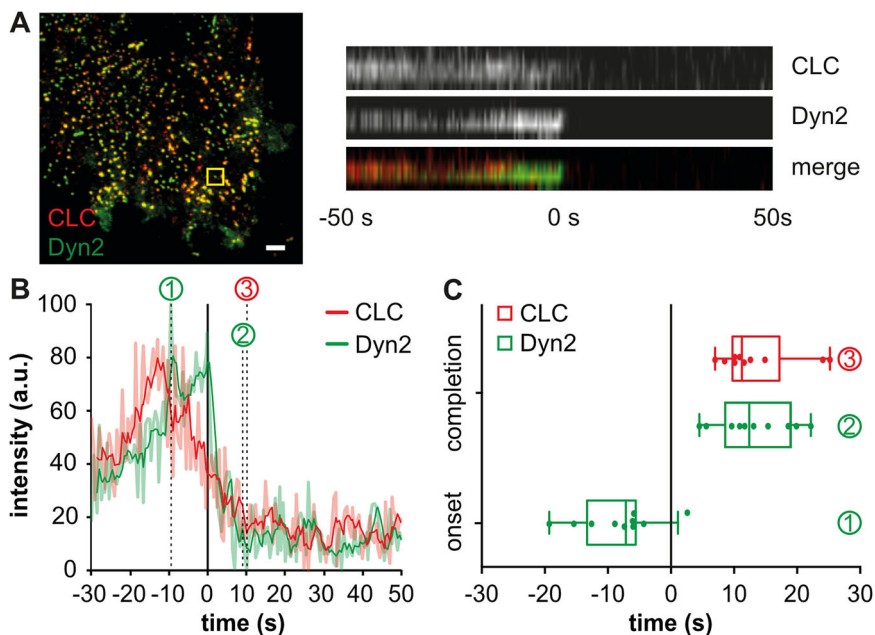

**Figure EV1.  Kinetics of dynamin recruitment during CME.**

(**A**) HeLa ATCC cells were co-transfected with mRFP-clathrin light chain (CLC) and EGFP-dynamin 2 (Dyn2). Cells were imaged by live-cell TIRF-M. Movies were acquired with 0.5 Hz frame rate for 5 min. CME events denoted by CLC signal loss were identified manually after background subtraction and filtering. The yellow box indicates the CME event shown as kymograph. Scale bar is 2 µm. (**B**) Plotted are the intensity profiles of CLC and Dyn (light red/green) as well as moving averages (intense red/green). Note that due to its additional role in vesicle maturation, Dyn2 was already present early during vesicle formation (Loerke et al, 2009; Taylor, Lampe and Merrifield, 2012). A second wave of recruitment was observed for scission and quantified in (**C**). (**C**) The onset of Dyn2 recruitment for scission (1) relative to the half-time of CLC loss from the cell surface ($t = 0$) as well as the timepoint of the completion of CLC (3) and Dyn2 (2) signal loss were manually determined from intensity profiles. Data information: For (**C**), depicted are values from 10 profiles (from $n = 3$ biological replicates). Dots in the bar graphs indicate the individual endocytosis events. Box plots indicate the interquartile range (25th to 75th percentiles, box) and median, while the whiskers extend to the minimum and maximum values of all data points.

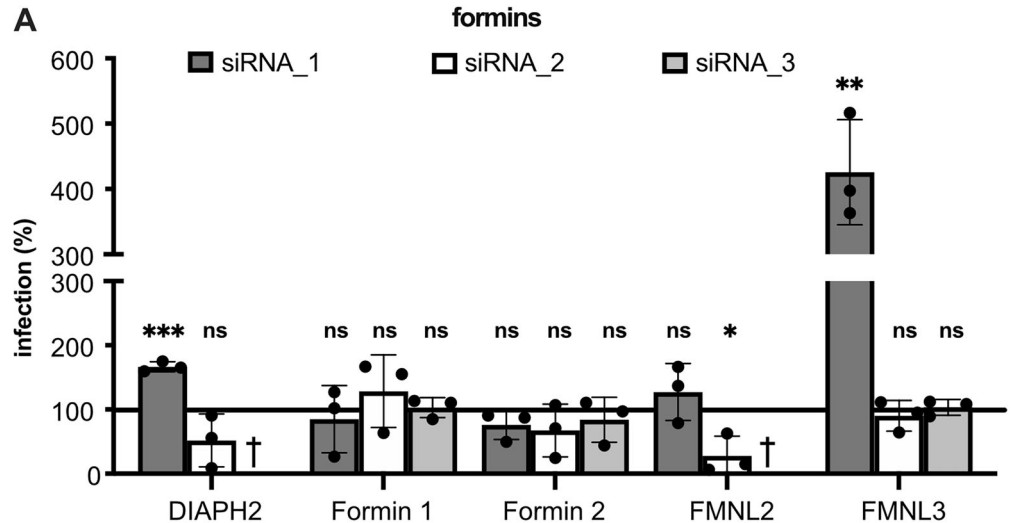

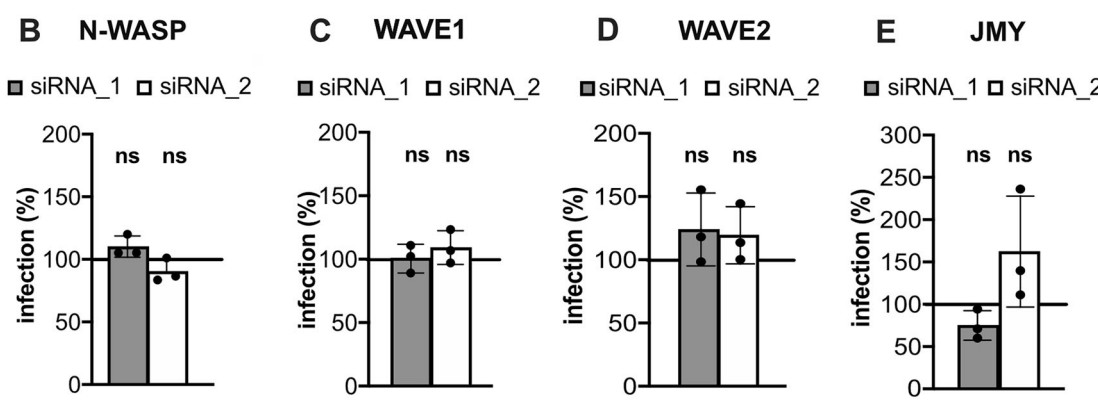

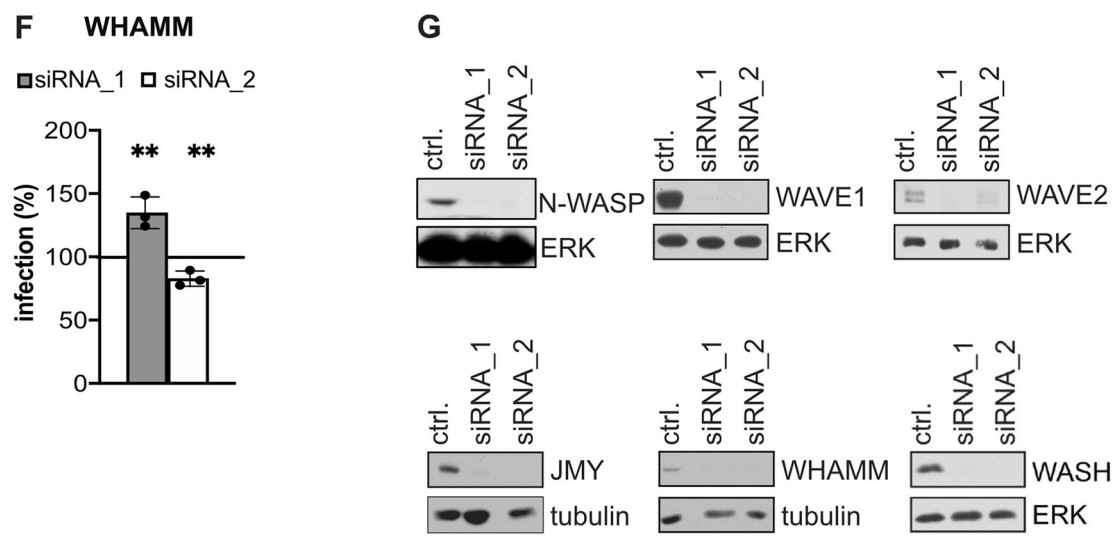

◀ **Figure EV2. Unbranched actin polymerization regulated by formins, nucleation-promoting factors N-WASP, WAVE, WHAMM and JMY are dispensable for HPV16 infection.**

(A) HeLa Kyoto cells were depleted of individual formins and infected with HPV16-GFP. Infection was scored 48 h p.i. by automated microscopy and normalized to ctrl., siRNAs that reduced cell numbers by more than 80% were considered cytotoxic (†) and excluded from the analysis. Dots in the bar graphs indicate the replicates. (B–F) HeLa Kyoto cells were infected with HPV16-GFP after siRNA mediated depletion of N-WASP (B), WAVE1 (C), WAVE2 (D), JMY (E), WHAMM (F). Infection was scored 48 h p.i. by automated microscopy, normalized to ctrl. ($n = 3$ biological replicates). Dots in the bar graphs indicate the replicates. (G) Protein expression levels were determined by Western blotting against the siRNA target proteins. Data information: For (A–F) data are represented as mean ± SD of $n = 3$ biological replicates. For all quantifications, statistical significance was assessed by Student's $t$ test (*$P \le 0.05$, **$P \le 0.01$, ***$P \le 0.001$, ****$P \le 0.0001$, ns = not significant). In (A), significance values in comparison to siRNA ctrl.: DIAPH2: siRNA_1: $P = 0.0001$, siRNA_2: $P = 0.1121$; Formin 1: siRNA_1: $P = 0.6476$, siRNA_2: $P = 0.4332$, siRNA_3: $P = 0.7612$; Formin 2: siRNA_1: $P = 0.1417$, siRNA_2: $P = 0.2425$, siRNA_3: $P = 0.4725$; FMNL2: siRNA_1: $P = 0.3478$, siRNA_2: $P = 0.0149$; FMNL3: siRNA_1: $P = 0.0022$, siRNA_2: $P = 0.5163$, siRNA_3: $P = 0.6605$. In (B), significance values in comparison to siRNA ctrl.: N-WASP: siRNA_1: $P = 0.0672$, siRNA_2: $P = 0.247$. In (C), significance values in comparison to siRNA ctrl.: WAVE1: siRNA_1: $P = 0.9101$, siRNA_2: $P = 0.3006$. In (D), significance values in comparison to siRNA ctrl.: WAVE2: siRNA_1: $P = 0.2236$, siRNA_2: $P = 0.2144$. In (E), significance values in comparison to siRNA ctrl.: WHAMM: siRNA_1: $P = 0.0087$, siRNA_2: $P = 0.0078$. In (F), significance values in comparison to siRNA ctrl.: JMY: siRNA_1: $P = 0.069$, siRNA_2: $P = 0.1747$.

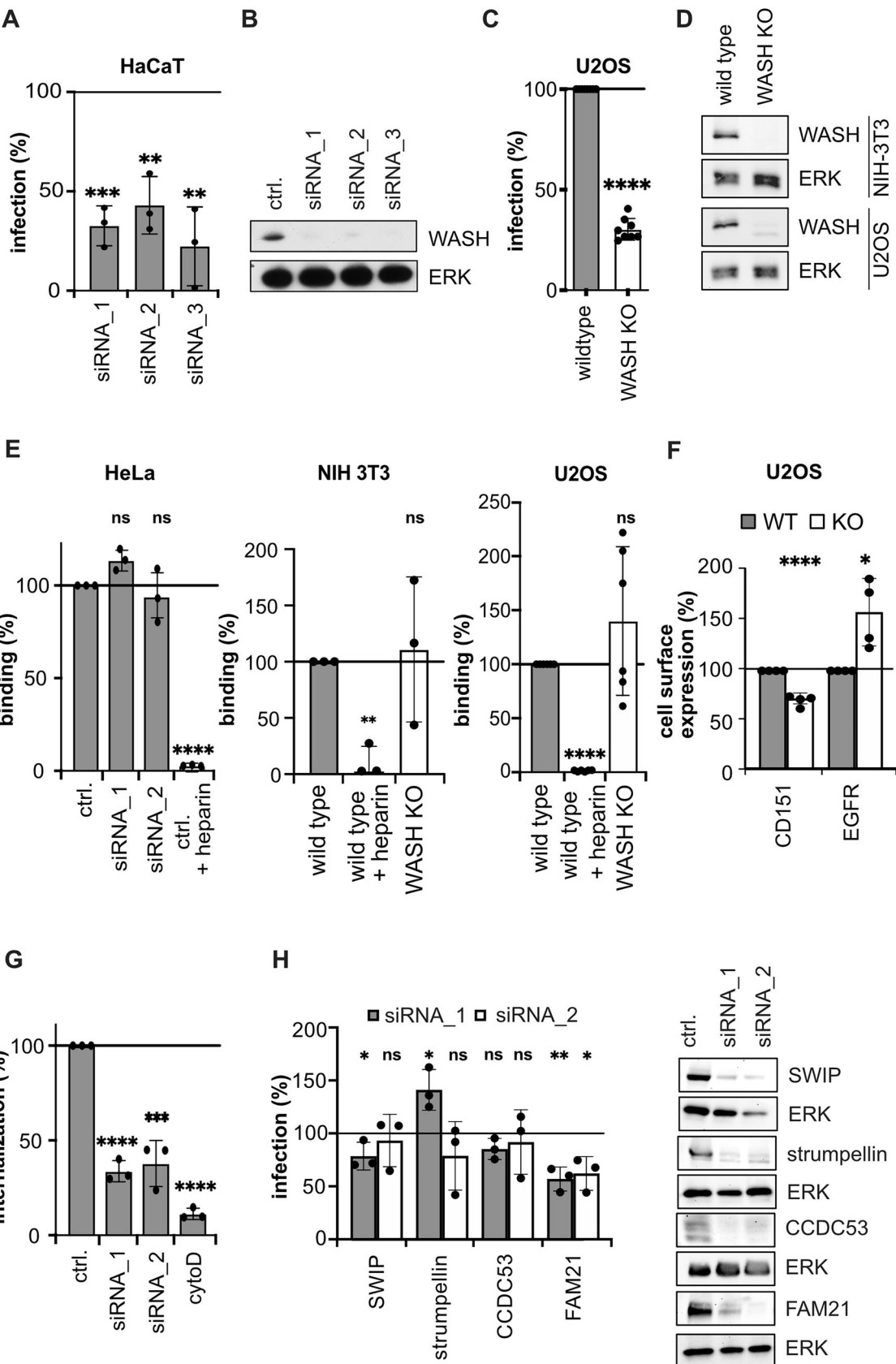

◄

**Figure EV3. Further characterization and analysis of WASH perturbation.**

(A) HaCaT cells were depleted of WASH by RNAi and subsequently infected with HPV16-GFP for 48 h. Infection levels were analyzed by automated microscopy and normalized to siRNA ctrl. (B) WASH depletion in HaCaT of (A) was confirmed by immunoblotting against WASH. (C) U2OS wild-type and WASH KO cells were infected with HPV16-GFP. Infection was analyzed 48 h p.i. by flow cytometry and normalized to wild-type cells. (D) WASH CRISPR/Cas9 KO in NIH-3T3 and U2OS KO cells was confirmed by immunoblotting against WASH in comparison to WT cells. (E) HPV16-AF488 was bound to Hela Kyoto cells transfected with ctrl. or WASH targeting siRNAs, NIH-3T3 or U2OS wild-type and WASH KO cells as indicated. Virus pre-incubated with 1 mg/ml heparin was used as a non-binding control. At 2 h p.i., virus binding was measured by flow cytometry, or in case for U2OS cells by automated microscopy. Values are shown as relatives to wild-type cells. (F) U2OS wild-type and WASH KO cells were detached by EDTA treatment. Extracellular epitopes of CD151 and EGFR were immunostained and cell surface levels were determined as the geometric mean intensity by flow cytometry. Values were normalized to wild-type cells. (G) HPV16-AF594 was bound to Hela Kyoto cells transfected with ctrl. or WASH targeting siRNAs and allowed to enter for 6 h. As control inhibiting uptake, cytochalasin D was used (10 µg/ml). The fluorescence of extracellular virus was quenched using trypan blue, and the fluorescence intensity of intracellular virus was analyzed by flow cytometry. Values are shown as relatives to ctrl. siRNA transfected cells. (H) HeLa Kyoto cells were depleted of SHRC proteins and infected with HPV16-GFP. Infection levels were analyzed by automated microscopy and normalized to ctrl. Protein levels were determined by Western blotting. Data information: In (A, C, E–H) data are represented as mean ± SD. For all quantifications, statistical significance was assessed by Student's $t$ test excepting (E, HeLa), where a Welsh's $t$ test has been used (*$P \leq 0.05$, **$P \leq 0.01$, ***$P \leq 0.001$, ****$P \leq 0.0001$, ns = not significant). In (A), data of $n = 3$ biological replicates. Significance values in comparison to siRNA ctrl.: siRNA_1: $P = 0.0003$, siRNA_2: $P = 0.0024$, siRNA_3: $P = 0.0025$. In (C), data of $n = 8$ biological replicates. Significance value in comparison to wild-type ctrl.: $P < 0.0001$. In (E), data of $n = 3$ excepting U2OS cells with $n = 6$ biological replicates. Significance values in comparison to ctrl.: HeLa cells (siRNA ctrl.): siRNA_1: $P = 0.0514$, siRNA2: $P = 0.4077$, siRNA ctrl.+ heparin: $P < 0{,}0001$; NIH 3T3 cells (wild-type ctrl.): NIH 3T3 cells + heparin: $P = 0.0016$, NIH 3T3 WASH KO: $P = 0.7838$; U2OS cells (wild-type ctrl.): U2OS cells + heparin: $P < 0{,}0001$, U2OS WASH KO: $P = 0.1849$. In (F), CD151 KO: $P < 0{,}0001$, EGFR KO: $P = 0.0156$. In (G), data of $n = 3$ biological replicates. Significance values in comparison to ctrl.: siRNA_1: $P < 0.0001$, siRNA_2: $P = 0.0009$, ctrl. + CytoD: $P < 0.0001$. In (H), data of $n = 3$ biological replicates. Significance values in comparison to ctrl.: SWIP – siRNA_1: $P = 0.0474$, siRNA_2: $P = 0.6608$; Strumpellin - siRNA_1: $P = 0.0207$, siRNA_2: $P = 0.3192$; CCDC53 – siRNA_1: $P = 0.0621$, siRNA_2: $P = 0.6655$; FAM21 - siRNA_1: $P = 0.0028$, siRNA_2: $P = 0.0148$.

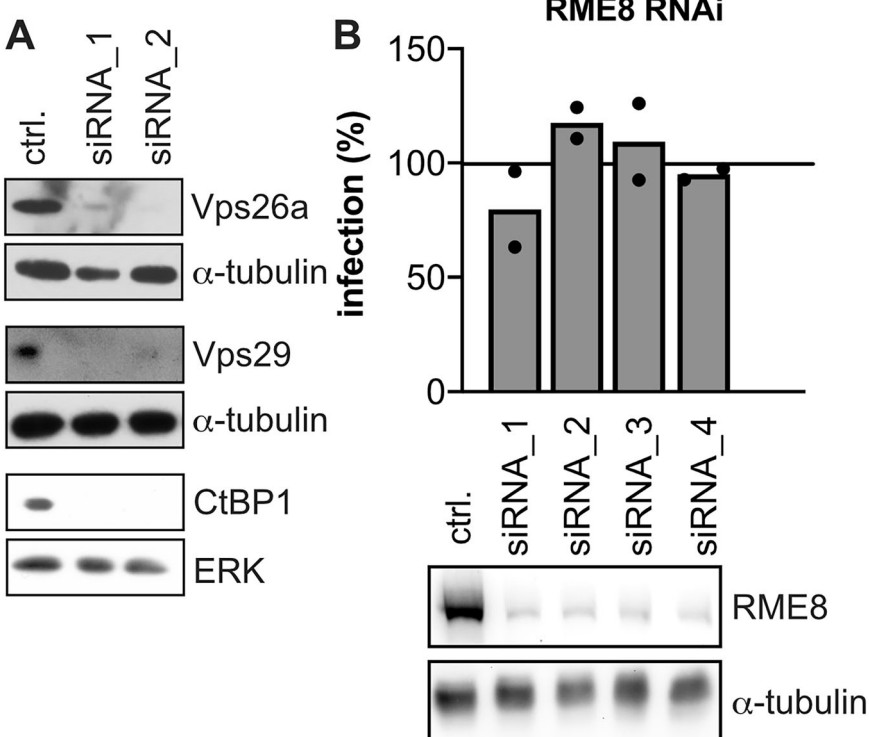

**Figure EV4. RME-8 is dispensable for HPV16 infection.**

(A) Depletion of Vps26a, Vps29, and CtBP1 by RNAi was confirmed by Western blotting. (B) RME8 was depleted from HeLa Kyoto cells by siRNA treatment and infected with HPV16-GFP. Infection was scored 48 h p.i. by automated microscopy and normalized to ctrl. Depicted is the mean of $n = 2$ biological replicates. Dots in the bar graph indicate the replicates. RME8 expression levels were analyzed by Western blotting.

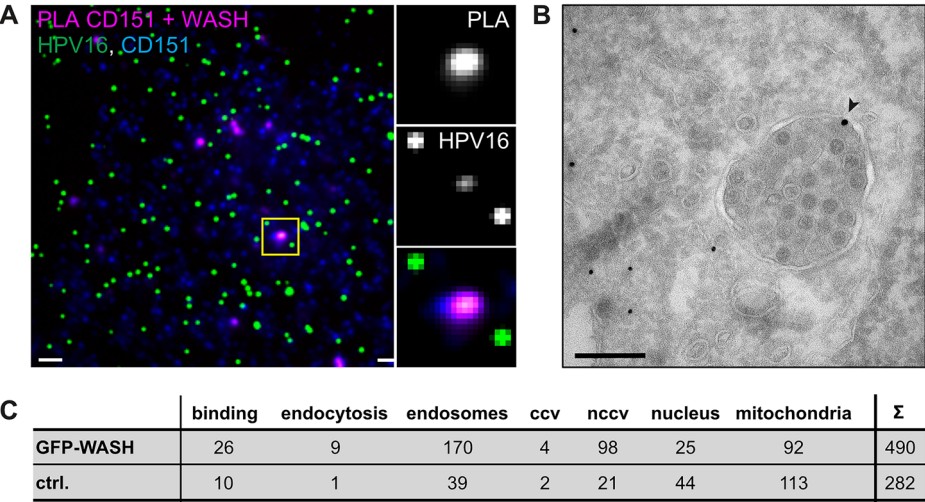

| **C** | binding | endocytosis | endosomes | ccv | nccv | nucleus | mitochondria | Σ |
|---|---|---|---|---|---|---|---|---|
| **GFP-WASH** | 26 | 9 | 170 | 4 | 98 | 25 | 92 | 490 |
| **ctrl.** | 10 | 1 | 39 | 2 | 21 | 44 | 113 | 282 |
| **fold enrichment** | 2.6 | 9 | 4.4 | 2 | 4.7 | 0.5 | 0.8 | - |

**Figure EV5. WASH is recruited to the plasma membrane.**

(A) HPV16-AF488 was bound to ECM. HaCaT cells transfected with the HA-CD151 were seeded on top. Close proximity between WASH and HA-CD151 was detected using a PLA and plasma membrane localization was analyzed by TIRF-M. Scale bars are 2 µm and 0.5 µm. (B) Specific detection of EGFP-WASH on endosomes by immunogold labeling. Scale bar is 200 nm. (C) The GFP antibody specificity was assessed by quantification of the localization to indicated cellular compartments. Depicted is the absolute count of gold particles in 50 randomly selected profiles of EGFP-WASH expressing and untransfected (ctrl.) cells from $n = 2$ biological replicates (ccv clathrin-coated intracellular vesicle, nccv non clathrin-coated intracellular vesicle).

