## [Peer Review File · EMBO Reports]

The actin nucleation promoting factor WASH facilitates clathrin-independent endocytosis of Human Papillomaviruses

Pia Brinkert, Lena Krebs, Pilar Samperio Ventayol, Lilo Greune, Carina Bannach, Cynthia Amakiri, Delia Bucher, Jana Kollasser, Petra Dersch, Steeve Boulant, Theresia Stradal, and Mario Schelhaas

Corresponding author(s): Mario Schelhaas (schelhaas@uni-muenster.de)

Review Timeline:

Submission Date:	21st Jan 25
Editorial Decision:	17th Feb 25
Revision Received:	26th May 25
Editorial Decision:	2nd Jul 25
Revision Received:	9th Sep 25
Accepted:	26th Sep 25

Editor: Achim Breiling / Martina Rembold

Transaction Report:

Dear Prof. Schelhaas,

Thank you for the submission of your manuscript to EMBO reports. I have now received the reports from the three referees that were asked to evaluate your study, which can be found at the end of this email.

As you will see, the referees think that these findings are of interest. However, they have several comments, concerns, and suggestions, indicating that a major revision of the manuscript is necessary to allow publication of the study in EMBO reports. As the reports are below, and all the referee concerns need to be addressed, I will not detail them here.

Given the constructive referee comments, I would like to invite you to revise your manuscript with the understanding that the concerns of the referees must be addressed in the revised manuscript and in a detailed point-by-point response. Acceptance of your manuscript will depend on a positive outcome of a second round of review. It is EMBO reports policy to allow a single round of revision only and acceptance of the manuscript will therefore depend on the completeness of your responses included in the next, final version of the manuscript.

- 1) a .docx formatted version of the final manuscript text (including legends for main figures, EV figures and tables), but without the figures included. Figure legends should be compiled at the end of the manuscript text.
- 2) individual production quality figure files as .eps, .tif, .jpg (one file per figure), of main figures and EV figures. Please upload these as separate, individual files upon re-submission.

- 4) a complete author checklist, which you can download from our author guidelines (<https://www.embopress.org/page/journal/14693178/authorguide>). Please insert page numbers in the checklist to indicate where the requested information can be found in the manuscript. The completed author checklist will also be part of the RPF.

- 5) that primary datasets produced in this study (e.g. RNA-seq, ChIP-seq, structural and array data) are deposited in an

appropriate public database. If no primary datasets have been deposited, please also state this in a dedicated section (e.g. 'No primary datasets have been generated and deposited'), see below.

The accession numbers and database should be listed in a formal "Data Availability" section that follows the model below. This is now mandatory (like the COI statement). Please note that the Data Availability Section is restricted to new primary data that are part of this study. This section is mandatory. As indicated above, if no primary datasets have been deposited, please state this in this section

Data availability

8) Regarding data quantification and statistics, please make sure that the number "n" for how many independent experiments were performed, their nature (biological versus technical replicates), the bars and error bars (e.g. SEM, SD) and the test used to calculate p-values is indicated in the respective figure legends (also for EV and Appendix figures). Please also check that all the p-values are explained in the legend, and that these fit to those shown in the figure. Please provide statistical testing where applicable. Please avoid the phrase 'independent experiment', but clearly state if these were biological or technical replicates. Please also indicate (e.g. with n.s.) if testing was performed, but the differences are not significant. In case n=2, please show the data as separate datapoints without error bars and statistics. See also: <http://www.embopress.org/page/journal/14693178/authorguide#statisticalanalysis>

9) Please add scale bars of similar style and thickness to microscopic images, using clearly visible black or white bars (depending on the background). Please place these in the lower right corner of the images themselves. Please do not write on or near the bars in the image but define the size in the respective figure legend.

10) Please also note our reference format:

12) We now use CRedit to specify the contributions of each author in the journal submission system. CRedit replaces the author contribution section. Please use the free text box to provide more detailed descriptions and do NOT provide your final manuscript text file with an author contributions section. See also our guide to authors: <https://www.embopress.org/page/journal/14693178/authorguide#authorshipguidelines>

13) All Materials and Methods need to be described in the main text using our 'Structured Methods' format, which is required for

all research articles. According to this format, the Methods section should include a Reagents and Tools Table (listing key reagents, experimental models, software, and relevant equipment and including their sources and relevant identifiers), uploaded as separate file, and a Methods section in which we encourage the authors to describe their methods using a step-by-step protocol format with bullet points, to facilitate the adoption of the methodologies across labs. More information on how to adhere to this format as well as downloadable templates (.doc) for the Reagents and Tools Table can be found in our author guidelines (section 'Structured Methods'):

Please include the siRNA information (Supplemental Table S1) into the Reagents and Tools Table.

14) Please order the sections like this, using these names:

Title page - Abstract - Keywords - Introduction - Results - Discussion - Methods - Data availability section - Acknowledgements (including the funding information) - Disclosure and Competing Interests Statement - References - Figure legends - Expanded View Figure legends

15) Please make sure that all the funding information is also entered into the online submission system and that it is complete and similar to the one in the acknowledgement section of the manuscript text file.

16) Please add a paragraph titled 'Biosafety' to the methods section gathering all information on where and how biosafety-relevant experiments with microbes were performed and that these were approved, and by whom (institution, government).

I look forward to seeing a revised form of your manuscript when it is ready.

Yours sincerely,

Achim Breiling
Sebior Editor
EMBO Reports

Referee #1:

The authors follow up on previous research further defining the unique endocytic path HPV16 utilizes for viral entry. Utilizing a variety of microscopic and EM, along with infectivity, methods, the authors show that the route HPV16 takes for endocytosis is clathrin-independent and distinct from micropinocytosis, confirming previous results. During this study, the authors discover that (branched) actin polymerization facilitates HPV16 endocytosis and that this actin polymerization is mediated by the nucleation factor WASH. Furthermore, the authors find that the role of WASH in HPV16 endocytosis is unique from its other defined roles at endosomes. This model was validated by the authors using inhibitor and knock-out (with supplementation) approaches. This manuscript is suitable for publication in EMBO and will be of interest to the HPV entry field. I only have a few minor points that need addressing first.

First, the Supplemental movies 1-3 were unplayable. I am unsure if this is an issue on my end or on EMBO's end.

Second, in lines 601-602, there's a mistype, it should read: "E-G) EGFP, WASH-YFP or the WASH mutants dWCA (E), K220R (F) or Y141F/E (G).

Third, minor points:

Line 267: Get rid of the reference to the (nonexistent) Figure 4(H).

Line 307: "(Change Figure S4F)" to "(Figure S4E)".

Get rid of the extra space in lines 385-394.

Line 458: Change "How WASH would be recruited to the plasma membrane, remains unclear" to "It remains unclear how WASH would be recruited to the plasma membrane."

Line 554: Remove the extra period in "6 h.p.i."

Line 561: Remove the extra period in "1 h.p.i."

Line 594: Remove the extra period in "6 h.p.i."

Line 611: Remove the extra period in "1 h.p.i."

Line 624: Remove the extra period in "6 h.p.i."

Line 879: Change "-ra bbit" to "-rabbit".

Line 986: Remove the extra period in "6 h.p.i."

Fix spacing in text in line 1275.

Referee #2:

The manuscript by Pia Brinkert et al., describes the mechanism of entry of the human Papillomavirus and reveals that it involves the actin nucleation promoting factor WASH and the generation of branched filamentous actin.

This highlights a novel role for WASH in endosome formation, distinct from its role in membrane budding during sorting and maturation of the compartments. The experiments are well performed. There are, however, some aspects to be clarified in order to strengthen the conclusions on the mechanism of internalization of the human Papillomavirus.

- To exclude a role on viral particles binding to the cell surface, the authors need to verify that virus binding is not affected in the various experimental conditions, in particular because WASH could play a role on the sorting and receptor delivery at the plasma membrane. Such control experiments are presented in figure S4D in NIH3T3 cells, but what about all the other experiments/ cell lines used?

- The authors assess cargo internalization by looking at the fluorescence from pHrodo-labeled HPV16, but there are some steps in between endocytosis and delivery to acidic endosomal compartments that could be potentially be affected by Arp3 depletion (figure 3D). Same comment for Figure 4B.

- in the same line, the authors assess HPV16 internalization by looking at the percentage of infected cells after a long time (48 hpi) and the observed phenotype might not be entirely explained by an effect on virus uptake. In addition, the fact that the virus is a pseudovirus is only mentioned in the methods and is not made clear in the main text.

Would it be possible to monitor endocytosis in a more direct manner in the experiments targeting the major molecular players?

- In the initial version of the preprint, there was a role for the retromer machinery (SNX2) that has been taken out of the manuscript. Why?

- Why is macropinocytosis presented in the final scheme? It is not the major focus of the results nor the discussion and the global scheme could be used in a more general review on endocytic pathways.

Minor concerns:

- Figure 1B: To improve clarity, the authors could add the condition "no obvious structure" (20-25% of viral particles according to the text and the legend) to the graph, along with a representative image.

- Titles in the Results part: should be in the present tense?

- Figure S3: a western blot to check the depletion of N-WASP is missing.

- It is confusing when the order of the panels in a given figure is not the same as the order in which the panels are referred to in the text (Figure 5D-E cited in the text before 5A-C, for example).

- Figure 5D: The endocytic pits seem bigger in the WASH KO cells compared to WT cells. Is that a relevant information?

- There are problems with the numbering of both Figure 4 and Figure S4: the letters of the panels do not always correspond to what is described in the text or in the legend (there are references in the text and legend to a Figure 4H that does not exist, description of Figure S4E in the legend does not correspond to the actual panel, etc.).

Clarification of the methods:

- Figure 1C, Figure 3A-C, Figure 4A,C-G: Could the authors mention in the legends of these figures that infection was assessed at 48 hpi? Why was the infection sometimes assessed by microscopy and sometimes by flow cytometry? Do the two methods yield similar results (in terms of percentage of infected cells)?
- The authors state in the Methods section that HPV16 was added to result in 20% of infected cells in the infection experiments. What dose of virus is that?
- There are many cell lines used in the manuscript (HeLa Kyoto vs. HeLa ATCC vs. NIH3T3 vs. U2OS). Is there any explanation for these various options?
- In the experiments where the virus internalization is monitored by TIRF-M: the virus seems to have been added on top of the cells... how can the entry then be visualized by TIRF-M?
- it would be good to show the results as bar charts with individual data points.

Referee #3:

Brinkert et al report that HPV16 enters cells via uncoated pits that required WASH protein at a late stage of endocytosis. They report this as a non-canonical function of WASH. Supporting evidence includes loss-of-function studies of proteins that are typically found in complex with WASH, which resulted in no change to HPV16 infection. WASH protein involvement in HPV infection in a non-canonical role was also evidenced by amino acid substitutions of WASH that affected its canonical functions, but not HPV infection. The involvement of WASH in HPV endocytosis is not only a major finding for HPV biology, but also for the understanding of actin nucleation promotion factors. Major strengths of this paper include the kymographic data, which convincingly depict the temporal relationship between signals, such as HPV and actin at the time of endocytosis in figure 1. The distinction between WASH involvement in HPV endocytosis and its canonical function in cells is well-established. Important shortcomings of this manuscript come in the form of seemingly missing data (NIH-3T3 infection that is referred to on line 265-267) and numerous textual errors. However, these can be ameliorated. For this manuscript to be considered further, these comments should be addressed:

Major:

- Figure 1A - the CME cargo is not defined. Is HPV uptake being compared to basally occurring CME? Is the uptake in question in this paper stimulated by HPV, or is HPV exploiting a process that would be active regardless of its presence?
- Why are the images in 1A (top row, and bottom right) repeated in 1B?
- Figure 2A - the immunogold labelling of actin is nearby the neck of the vesicles depicted, but it is not convincing, because there is also labelling not near the neck of the vesicles. It makes it seem like there just happens to be labelling at the site of interest. Ultimately it is difficult to correlate the immunogold labelling with the neck of the vesicle from these images. Perhaps better definition of "at the neck of constricted endocytic pits" will strengthen the message of these images.
- Figure 4A - 2 siRNAs against the same target are used; the functional consequence of RNAi on infection is similar between both siRNAs for HPV, but not VV. The conclusion drawn is that there is a mild effect on VV infection, but this must be based on the single siRNA that resulted in statistical significance. A Third siRNA could change this conclusion in either direction.
- Results refer to Figure 4H but there is no panel H. Furthermore, the infection data in NIH3T3 is absent.

Minor:

Intro:

Line 69: allows "us" or "researchers"

Results:

-The figure legend explains that the images in 1A depict endocytic pits, but it would be helpful to point out the virus particles that are in the images.

Figure 1D - it is confusing that the control has 1 virus particle in 1 invagination, while the cytoD image has numerous particles. How does cytoD treatment result in an increased number of particles per entry location?

Line 293 Crispr/Cas9

Line 299 delete "the" or "its"

Line 307 should refer to S4E

Line 314 should refer to figure 4E-G

Line 333 should refer just Figure S4D. S4E regards WASH complex members (if this has to do with recycling or binding, it should be made clear)

Line 348 I think a period is necessary after the word 'polymerization'

Figure 5 is described in the results starting with D and E, then onto A, B & C. Can either the panels or the text be reordered?

Figure 5A could include more context to better understand the image. The WASH signal doesn't appear very specific, but rather, there appears to be quite a bit of background

Figure 5C I would have expected the anti-actin antibody to label actin in close proximity to the late pit, at least, especially with regard to figure 2B.

Discussion:

Line 426-427: there are some highlighted words (this may just be in this proof, and not in the final version)

Materials & Methods:

-Could you confirm the actin antibody used for this study? A Google search for different combinations of keywords BRICS mouse anti actin (4F7) bsbs300470 did not yield results.

Point-by-point response to the reviewers' concerns

First, we would like to thank all the reviewers for their constructive criticism that allowed us to strengthen our manuscript.

For your notice, we have adapted the Figures to EMBO reports style and made the following changes to the figures:

Figure 5: rearranged the panels to match the text in the Results section.

Supplementary Figure 1: now Appendix Figure 1

Supplementary Figure 2: now Figure EV1

Supplementary Figure 3: now Figure EV2

Supplementary Figure 4: now Figure EV3

Supplementary Figure 5: now Figure EV4

Supplementary Figure 6: now Appendix Figure 2

Supplementary Movie 1: now Movie EV1

Supplementary Movie 2: now Movie EV2

Supplementary Movie 3: now Movie EV3

New experimental data has been added to Figure EV3 and includes binding data upon WASH silencing and knockdown, infection data of HaCaT cells upon WASH silencing, new internalization data measuring particle uptake.

Source data has been provided, and can be accessed for review purposes using the following link:

<https://www.ebi.ac.uk/biostudies/studies/S-BSST2069?key=c2a2031e-58e6-4e8e-8ee8-732c49edeac9>

Referee #1:

The authors follow up on previous research further defining the unique endocytic path HPV16 utilizes for viral entry. Utilizing a variety of microscopic and EM, along with infectivity, methods, the authors show that the route HPV16 takes for endocytosis is clathrin-independent and distinct from micropinocytosis, confirming previous results. During this study, the authors discover that (branched) actin polymerization facilitates HPV16 endocytosis and that this actin polymerization is mediated by the nucleation factor WASH. Furthermore, the authors find that the role of WASH in HPV16 endocytosis is unique from its other defined roles at endosomes. This model was validated by the authors using inhibitor and knock-out (with supplementation) approaches. This manuscript is suitable for publication in EMBO

and will be of interest to the HPV entry field. I only have a few minor points that need addressing first.

We would like to thank the reviewer for the positive assessment of our efforts.

First, the Supplemental movies 1-3 were unplayable. I am unsure if this is an issue on my end or on EMBO's end.

The movies are playable on our side, and will be checked by the EMBO reports team after submission of the revised manuscript. This will hopefully rectify the issue.

Second, in lines 601-602, there's a mistype, it should read: "E-G) EGFP, WASH-YFP or the WASH mutants dWCA (E), K220R (F) or Y141F/E (G).

We thank the reviewer for noting this, we have corrected the mistype accordingly.

Third, minor points:

Line 267: Get rid of the reference to the (nonexistent) Figure 4(H).

We have corrected the reference from 'Fig. 4H' to 'Fig. 4F and 4G'.

Line 307: "(Change Figure S4F)" to "(Figure S4E)".

We have changed the reference to Fig. S4E.

Get rid of the extra space in lines 385-394.

Removed according to the reviewer's suggestion.

Line 458: Change "How WASH would be recruited to the plasma membrane, remains unclear" to "It remains unclear how WASH would be recruited to the plasma membrane."

Changed according to the reviewer's suggestion.

Line 554: Remove the extra period in "6 h.p.i."

Line 561: Remove the extra period in "1 h.p.i."

Line 594: Remove the extra period in "6 h.p.i."

Line 611: Remove the extra period in "1 h.p.i."

Line 624: Remove the extra period in "6 h.p.i."

We removed the extra period in all instances.

Line 879: Change "-ra bbit" to "-rabbit".

Changed.

Line 986: Remove the extra period in "6 h.p.i.".

Extra period was removed.

Fix spacing in text in line 1275.

Fixed.

Referee #2:

The manuscript by Pia Brinkert et al., describes the mechanism of entry of the human Papillomavirus and reveals that it involves the actin nucleation promoting factor WASH and the generation of branched filamentous actin.

This highlights a novel role for WASH in endosome formation, distinct from its role in membrane budding during sorting and maturation of the compartments. The experiments are well performed. There are, however, some aspects to be clarified in order to strengthen the conclusions on the mechanism of internalization of the human Papillomavirus.

- To exclude a role on viral particles binding to the cell surface, the authors need to verify that virus binding is not affected in the various experimental conditions, in particular because WASH could play a role on the sorting and receptor delivery at the plasma membrane. Such control experiments are presented in figure S4D in NIH3T3 cells, but what about all the other experiments/ cell lines used?

Thank you for this comment. We obviously agree that a potential role of the WASH complex in the putative recycling of viral receptors through retromer-associated functions needs to be assessed. This is why we already provided experimental evidence to address this issue In the previous version of the manuscript also beyond a mere binding control. To experimentally address the reviewer's comments, we added virus binding experiments for HeLa cells upon WASH knockdown and U2OS KO cells, and, both of which indicate that virus binding is unaffected (Fig. EV3).

In corroboration, we already presented direct and indirect evidence that WASH perturbation does not impact virus infection through receptor recycling. In the

previous Fig. S4D (now Fig. EV3E) we exclude a defect in binding to NIH3T3 WASH KO cells. In addition, we tested cell surface expression of secondary receptor candidates in U2OS WASH KO cells (now Fig. EV3F). More indirectly, we show that WASH KO results in an increase of virus particle-filled, fully formed endocytic pits in NIH3T3 cells indicating that the defect is unlikely due to binding or endocytic pit formation but rather due to a failed scission event (now Fig. 5A). Spatiotemporal association of viruses with endocytic pits and endocytosing virus indicates a functional association of WASH at the plasma membrane (Fig. 5C-E) in HeLa cells. The ability of the WASH ubiquitination mutant incapable of activation at endosomes to rescue infectivity in NIH3T3 WASH KO cells indicates that a different function than recycling via retromer is responsible for failed infection upon loss of WASH (Fig. 4F). Similarly, the inability of depleting several members of the WASH regulatory complex at endosomes (excepting FAM21) to interfere with virus infection supports this notion (Fig. EV3H).

Thus, the cumulative evidence all supports a role of WASH during virus endocytosis rather in uptake than binding/recycling of receptors.

- The authors assess cargo internalization by looking at the fluorescence from pHrodo-labeled HPV16, but there are some steps in between endocytosis and delivery to acidic endosomal compartments that could be potentially be affected by Arp3 depletion (figure 3D). Same comment for Figure 4B.

We thank the reviewer for this comment. Intracellular transport has been shown to be microtubule-dependent (Schelhaas *et al.*, 2012). Nevertheless, it is conceivable that HPV entry involves an additional actin-mediated intracellular transport step. Yet, previous work from us already demonstrated that cytochalasin D-suppressed actin polymerization interferes with particle uptake but not binding (Schelhaas *et al.*, 2012). In the respective experiment, virus translocation across the plasma membrane of HeLa cells was directly tested by quenching the fluorescence of extracellular virions and thereby assessing the accumulation of intracellular virions irrespective of their delivery to endosomes. Upon cytochalasin D treatment, translocation was entirely blocked. Hence, even if all actin polymerization is blocked, the primary defect relates to uptake of virions but not intracellular actin-mediated transport. Similar results and conclusions exist for HPV33 (Selinka *et al.*, 2002). Since depletion of Arp3 interferes with branched actin polymerization, it is most plausible that Arp3 depletion also affects primarily virus uptake. We have now mentioned this more clearly and referenced the previous work. These experiments do not rule out that Arp2/3-mediated actin polymerization might to some degree promote intracellular transport in addition to uptake. Hence, we do acknowledge this in the revised manuscript (see lines 236-239).

To verify this question for WASH itself (Fig. 4B), we used the quenching assay upon WASH silencing in HeLa cells. Our data showed that HPV particle uptake itself is reduced upon WASH KD (Fig. EV3G) confirming that WASH aids endocytosis. As for

actin polymerization, we cannot formerly rule out additional contributions of WASH to intracellular transport, which we now mention in the results and discussion (lines 272-274, 397-398).

- in the same line, the authors assess HPV16 internalization by looking at the percentage of infected cells after a long time (48 hpi) and the observed phenotype might not be entirely explained by an effect on virus uptake.

The reviewer is obviously correct: reduced infection on its own does not indicate that the perturbed infectivity is entirely caused by defective uptake in our knockdown and knockout experiments. This is why we used several lines of evidence to conclude that branched actin polymerization mediated by WASH facilitates endocytosis. Our previous and new data testing infectivity, binding, and internalization upon WASH perturbation in conjunction with the observed spatiotemporal association of polymerizing actin and WASH to internalizing virus render the proposed function of WASH-mediated actin polymerization to primarily aid scission the most plausible interpretation. In line with the reviewer's concern, we do agree that we cannot rule out additional effects of WASH and actin polymerization on virus infection after virus endocytosis. To make this clearer, we now explicitly mention as mentioned above.

In addition, the fact that the virus is a pseudovirus is only mentioned in the methods and is not made clear in the main text.

Thank you for mentioning this. We added a dedicated sentence in the introduction to mention the use of pseudoviruses in lines 120-123.

Would it be possible to monitor endocytosis in a more direct manner in the experiments targeting the major molecular players?

We have now added a more direct internalization assay to show that WASH depletion interferes with internalization in HeLa cells (Fig. EV3G), and refer to published data on actin polymerization. We think these additions prompted by the reviewer's comment strengthened our conclusions.

- In the initial version of the preprint, there was a role for the retromer machinery (SNX2) that has been taken out of the manuscript. Why?

We have removed the data on SNX2, as our initial assumptions on a potential role of SNX2 from a previous, and now corrected BioRxiv submission are no longer supported. We based these assumptions on RNAi-mediated depletion. However, as shown in this manuscript, well-characterized double knockouts of SNX1/SNX2 or SNX5/6 do not significantly affect infectivity. We thus suspect that off-target effects of the SNX2-targeting siRNAs interfered with infection. Alternatively, the putative function of SNX2 during virus endocytosis can be compensated upon knockout.

Thus, we decided to remove this data from the current manuscript, and mentioned that the SNX BAR dimer is not essential for endocytosis as evidenced by the knockout data. However, we aim to resolve the issue of SNX2 in our future work.

- Why is macropinocytosis presented in the final scheme? It is not the major focus of the results nor the discussion and the global scheme could be used in a more general review on endocytic pathways.

We added macropinocytosis to our final scheme, since we concluded in our previous work that the endocytic uptake pathway of HPV16 would be 'macropinocytosis-like', as it involves the sodium/proton exchanger and actin-polymerization (Schelhaas et al., 2012). We now specifically mention this in the discussion (lines 393-396) to correct our previous notion. We decided not to discuss our results in the context of all different endocytic mechanisms proposed in the literature, as there is no general consensus in the field on, for instance, the diversity of cholesterol-dependent, clathrin-independent mechanisms. We thus focused on the best studied mechanism, clathrin-independent endocytosis, and micropinocytosis, which has been assumed to be the closest relative to HPV endocytosis.

Minor concerns:

- Figure 1B: To improve clarity, the authors could add the condition "no obvious structure" (20-25% of viral particles according to the text and the legend) to the graph, along with a representative image.

We have added this data set as requested including a representative image. Of note, in the previous quantification, membrane folds or other unclear structures that could be due to sample preparation have been omitted from the quantification. We now added them to the quantification to avoid introducing a bias.

- Titles in the Results part: should be in the present tense?

We amended the titles in the results part according to the reviewer's comment.

- Figure S3: a western blot to check the depletion of N-WASP is missing.

Thank you for noting this embarrassing oversight, we have added the (obviously existing) data as requested.

- It is confusing when the order of the panels in a given figure is not the same as the order in which the panels are referred to in the text (Figure 5D-E cited in the text before 5A-C, for example).

We have changed Figure 5 to reflect the order in which the panels are mentioned in the text.

- Figure 5D: The endocytic pits seem bigger in the WASH KO cells compared to WT cells. Is that a relevant information?

The endocytic pits in the image of WASH KO cells indeed appear slightly bigger. As previously observed, virus containing pits vary somewhat in size and form in unperturbed cells (Schelhaas *et al.*, 2012). During analysis, the variation of pit size between WT and KO cells was measured and found to be insignificant. Hence, we do not consider this a relevant issue. This is now mentioned in the figure legend.

- There are problems with the numbering of both Figure 4 and Figure S4: the letters of the panels do not always correspond to what is described in the text or in the legend (there are references in the text and legend to a Figure 4H that does not exist, description of Figure S4E in the legend does not correspond to the actual panel, etc.).

We apologize for the confusion. We have now amended all references in the text and legend to reflect the actual figures.

Clarification of the methods:

- Figure 1C, Figure 3A-C, Figure 4A,C-G: Could the authors mention in the legends of these figures that infection was assessed at 48 hpi?

Certainly, we have added when infection was scored in the figure legends.

Why was the infection sometimes assessed by microscopy and sometimes by flow cytometry? Do the two methods yield similar results (in terms of percentage of infected cells)?

We have used microscopy and flow cytometry analysis of infection interdependently over the years and have not found any notable or significant differences between the methods. In other words, neither in terms of absolute nor relative data points, a difference in percentage of infected cells is observable. The methodology for analysis used in this manuscript simply reflects the availability of equipment or the preference of the researcher performing the experiment.

- The authors state in the Methods section that HPV16 was added to result in 20% of infected cells in the infection experiments. What dose of virus is that?

We have added the requested information in the methods section.

- There are many cell lines used in the manuscript (HeLa Kyoto vs. HeLa ATCC vs. NIH3T3 vs. U2OS). Is there any explanation for these various options?

In our hands, HeLa Kyoto and ATCC have always been equivalent in the outcome of a study, but are differently suitable for certain kinds of experimentation. While HeLa Kyoto cells are more easily susceptible to transfection, we prefer to use those cells for RNAi studies, as lower doses of siRNAs can be used and thus reduce the risk of off target effects. HeLa ATCC are more amenable to microscopy. The NIH3T3 and U2OS cells have been available to us as thoroughly characterized WASH KO cells. As we have shown that WASH silencing in HeLa cells perturbs infection and endocytosis of HPV16, the use of these different cell types confirmed our findings, provided evidence for a ubiquitous requirement of WASH in various cell types for this clathrin-independent endocytosis pathway, and allowed to easily extend our experimental strategy to electron microscopy and rescue analysis in a null background. We have now added also HaCaT cells as a keratinocyte cell line to cover all grounds.

- In the experiments where the virus internalization is monitored by TIRF-M: the virus seems to have been added on top of the cells... how can the entry then be visualized by TIRF-M?

The virus is small enough to diffuse underneath the cell. This strategy has been used successfully for a number of viruses (e.g. (Ewers *et al*, 2005; Johannsdottir *et al*, 2009; Mazel-Sanchez *et al*, 2023)), but obviously might be problematic for larger viruses such as vaccinia.

- it would be good to show the results as bar charts with individual data points.

We have added the individual data points to the graphs.

Referee #3:

Brinkert et al report that HPV16 enters cells via uncoated pits that required WASH protein at a late stage of endocytosis. They report this as a non-canonical function of WASH. Supporting evidence includes loss-of-function studies of proteins that are typically found in complex with WASH, which resulted in no change to HPV16 infection. WASH protein involvement in HPV infection in a non-canonical role was also evidenced by amino acid substitutions of WASH that affected its canonical functions, but not HPV infection. The involvement of WASH in HPV endocytosis is not only a major finding for HPV biology, but also for the understanding of actin nucleation promotion factors. Major strengths of this paper include the kymographic data, which convincingly depict the temporal relationship between signals, such as HPV and actin at the time of endocytosis in figure 1. The distinction between WASH involvement in HPV endocytosis and its canonical function in cells is well-established. Important shortcomings of this manuscript come in the form of seemingly missing

data (NIH-3T3 infection that is referred to on line 265-267) and numerous textual errors. However, these can be ameliorated. For this manuscript to be considered further, these comments should be addressed:

While the data were present in the figures, they were in some instances wrongly referenced. We do acknowledge that this - in combination with the textual errors - must have been annoying, and we sincerely apologize for the confusion.

Major:

-Figure 1A - the CME cargo is not defined. Is HPV uptake being compared to basally occurring CME?

Indeed, HPV uptake is compared to basally occurring CME, we now mention this in the figure legend and results.

- Is the uptake in question in this paper stimulated by HPV, or is HPV exploiting a process that would be active regardless of its presence?

This is an interesting question that – while not a focus of our study - we also have been wondering about:

The requirements for certain signal transduction events (e.g. growth factor signaling) for pit formation have been established by us and others (Bannach *et al*, 2020; Schelhaas *et al.*, 2012; Surviladze *et al*, 2012). While we would thus assume that endocytosis is stimulated by HPV (and likely cellular ligands), we (and others) have been unable to substantiate this notion. On the other hand, the cytochalasin D-mediated inhibition of actin polymerization leads to several viruses being located within the elongated tubular pits, which could be explained by sorting of viruses to entry sites or endocytic pits.

To experimentally address this issue is not trivial, particularly as sorting and induction are not mutually exclusive: Due to the asynchronous mode of virus internalization, where individual virus particles switch receptors and endocytose in a seemingly stochastic manner over many hours, typical biochemical assays to assess e.g. phosphorylation of effectors of virus-stimulated signaling yield no conclusive results (low signal above background). Also, assessing effector recruitment to virus particles at the plasma membrane only yields insignificant correlations at specific time points due to the asynchronous mode of internalization. To differentiate between local pit induction or lateral sorting to preexisting pits on the plasma membrane, one would have to analyze correlative live cell data and document the association of virus with a preexisting pit structure or live formation of such a structure at the site of a virus particle in significant numbers. With WASH we have identified the first potential candidate to use for such an analysis. Based on the existing data, however, we cannot conclude on this question. Given that only one or two entry events per movie can be observed due to the protracted virus uptake time course, and given that in a significant proportion of internalization events WASH and the virus are already correlated, when image acquisition started 1 h p.i., we estimate that we would need

to analyze at least a hundred additional movies to reach significance. Given the existing experimental hurdles, addressing this issue would be a major effort that we think would be beyond the scope of the current manuscript. As we agree on the importance of this question, we now added this issue to the discussion.

Of note, the identity of the secondary receptor remains elusive, so that we cannot analyze induction and internalization of this elusive receptor independently of a viral stimulus, which would indicate which cellular cargo this mechanism typically uses. As WASH localization to the plasma membrane is observable but infrequent, the current data would indicate that this pathway is active only to a minor extent under typical cell culture conditions.

-Why are the images in 1A (top row, and bottom right) repeated in 1B?

We simply repeated them as a visual aid for the reader, when displaying the quantitation. This may be particularly useful to appreciate what we refer to as actin networks or no clear structure in comparison.

-Figure 2A - the immunogold labelling of actin is nearby the neck of the vesicles depicted, but it is not convincing, because there is also labelling not near the neck of the vesicles. It makes it seem like there just happens to be labelling at the site of interest. Ultimately it is difficult to correlate the immunogold labelling with the neck of the vesicle from these images. Perhaps better definition of "at the neck of constricted endocytic pits" will strengthen the message of these images.

Our description of the immunogold labelling of actin appears indeed overly suggestive. We actually aimed to point out that there are instances, in which actin signals appeared to be clustered at the neck. This would be consistent with actin aiding in scission. Even much additional effort in quantification may be inconclusive, given that the labelling efficiency in immunogold electron microscopy is between 0.01 to 0.1. Thus, even with actin accumulation present, there would not be always a signal, whereas the reviewer is right to point out that the immunogold label could just happen to be at the 'preferred' site. Hence, we rephrased our description to reflect this (lines 184-187): *Consistent with a role of actin polymerization aiding vesicle scission, the presence of actin was detectable close to constricted endocytic pits in immunogold labelling EM (Fig. 2A).*

Strong evidence for actin polymerization facilitating endocytosis stems from our live cell imaging of endocytosis and actin polymerization dynamics, which indicates a burst of actin polymerization just before and during virus internalization.

-Figure 4A - 2 siRNAs against the same target are used; the functional consequence of RNAi on infection is similar between both siRNAs for HPV, but not VV. The conclusion drawn is that there is a mild effect on VV infection, but this must be based on the single siRNA that resulted in statistical significance. A Third siRNA could change this conclusion in either direction.

We have added additional RNAi data for HPV and VV, which shows that this third siRNA does not affect VV infection. We have amended the description accordingly.

-Results refer to Figure 4H but there is no panel H. Furthermore, the infection data in NIH3T3 is absent.

This is indeed a wrong reference to a non-existing figure panel. In fact, it should refer to Fig. 4F and G, in which infection of NIH3T3 and NIH3T3 WASH KO cells (including rescue experiments by ectopic overexpression of WASH and WASH mutants) are compared. We have corrected our mistake.

Minor:

Intro:

Line 69: allows "us" or "researchers"

We added 'researchers'.

Results:

-The figure legend explains that the images in 1A depict endocytic pits, but it would be helpful to point out the virus particles that are in the images.

As these are metal replicas of basal plasma membrane sheets and they are viewed from the luminal side of the cell, the viruses are not visible in the electron micrographs. The virus particles were fluorescently labeled and any structures were correlated with their fluorescent signal (depicted in the inset). We amended the description in the results and figure legend to clarify this better.

Figure 1D - it is confusing that the control has 1 virus particle in 1 invagination, while the cytoD image has numerous particles. How does cytoD treatment result in an increased number of particles per entry location?

This is again an excellent question. It relates to an earlier question of this reviewer whether endocytosis is induced by HPV16 or active in unstimulated cells. As explained above, we could rationalize that the presence of several virus particles in tubular pits upon cytoD treatment relates to a certain degree of virus sorting to preferred entry sites. Note that even in unperturbed cells, occasionally more than one virus can be found within one pit (Schelhaas *et al.*, 2012). We have now added a brief discussion of this issue stating that this question remains open (lines 467-473).

Line 293 Crispr/Cas9

We corrected the mistake.

Line 299 delete "the" or "its"

Again, we corrected the mistake.

Line 307 should refer to S4E

We now refer to the correct Fig. S4E.

Line 314 should refer to figure 4E-G

This is correct, and we changed the manuscript accordingly.

Line 333 should refer just Figure S4D. S4E regards WASH complex members (if this has to do with recycling or binding, it should be made clear)

This is correct. We have removed the reference to Fig S4E.

Line 348 I think a period is necessary after the word 'polymerization'

Thank you for noting this, we have inserted the period as requested.

Figure 5 is described in the results starting with D and E, then onto A, B & C. Can either the panels or the text be reordered?

The figure has been reordered to reflect the text.

Figure 5A could include more context to better understand the image. The WASH signal doesn't appear very specific, but rather, there appears to be quite a bit of background

We used cells with a rather low expression of WASH-GFP to avoid potential effects from ectopic expression. Hence, the WASH signal may be weak, but is clearly above background. Here, we provide a line intensity profile (left) of the virus associated WASH signal (right). In order to provide more context, we have now included an enlarged the field of view in the manuscript, and provide not only the merged image but single channel images as well.

Figure 5C I would have expected the anti-actin antibody to label actin in close proximity to the late pit, at least, especially with regard to figure 2B.

As we laid out in response to the first question, this is not necessarily what one could expect due to the low labeling efficiency during immunogold EM (0.01-0.1).

Discussion:

Line 426-427: there are some highlighted words (this may just be in this proof, and not in the final version)

Thank you for this observation. We have not observed the highlights in our word document, this must have been an error of some kind during PDF conversion.

Materials & Methods:

-Could you confirm the actin antibody used for this study? A Google search for different combinations of keywords BRICS mouse anti actin (4F7) bsbs300470 did not yield results.

We apologize for the unclear description. This antibody has been generated at the former antibody facility of the Braunschweig Integrated Centre for Systems Biology (BRICS) in collaboration with Brigitte Jockusch (Schroeder *et al*, 2009). As the original homepage link has been discontinued, we are providing a link for a contact (<https://www.tu-braunschweig.de/brics>). We have amended the description of the reagent accordingly.

References:

Bannach C, Brinkert P, Kühling L, Greune L, Schmidt MA, Schelhaas M (2020) Epidermal growth factor receptor and Abl2 kinase regulate distinct steps of Human papillomavirus type 16 endocytosis. *Journal of virology* 2
 Ewers H, Smith AE, Sbalzarini IF, Lilie H, Koumoutsakos P, Helenius A (2005) Single-particle tracking of murine polyoma virus-like particles on live cells and artificial membranes. *Proc Natl Acad Sci U S A* 102: 15110-15115
 Johannsdottir HK, Mancini R, Kartenbeck J, Amato L, Helenius A (2009) Host cell factors and functions involved in vesicular stomatitis virus entry. *Journal of virology* 83: 440-453

Mazel-Sanchez B, Niu C, Williams N, Bachmann M, Choltus H, Silva F, Serre-Beinier V, Karenovics W, Iwaszkiewicz J, Zoete V *et al* (2023) Influenza A virus exploits transferrin receptor recycling to enter host cells. *Proc Natl Acad Sci U S A* 120: e2214936120

Schelhaas M, Shah B, Holzer M, Blattmann P, Kühling L, Day PM, Schiller JT, Helenius A (2012) Entry of human papillomavirus type 16 by actin-dependent, clathrin- and lipid raft-independent endocytosis. *PLoS pathogens* 8: e1002657-e1002657

Schroeder U, Graff A, Buchmeier S, Rigler P, Silvan U, Tropel D, Jockusch BM, Aebi U, Burkhard P, Schoenenberger CA (2009) Peptide nanoparticles serve as a powerful platform for the immunogenic display of poorly antigenic actin determinants. *J Mol Biol* 386: 1368-1381

Selinka HC, Giroglou T, Sapp M (2002) Analysis of the infectious entry pathway of human papillomavirus type 33 pseudovirions. *Virology* 299: 279-287

Surviladze Z, Dziduszko A, Ozbun MA (2012) Essential roles for soluble virion-associated heparan sulfonated proteoglycans and growth factors in human papillomavirus infections. *PLoS pathogens* 8: e1002519-e1002519

Dear Prof. Schelhaas,

Thank you for the submission of your revised manuscript to our editorial offices. I have now received the reports from the two referees that I asked to re-evaluate the study, you will find below. Referee #1 was already very positive regarding the first version, and I consider his/her minor points as adequately addressed. As you will see, the other two referees now support the publication of your manuscript in EMBO reports. Referee #3 has a final suggestion to improve the discussion, I ask you to address in a final revised manuscript.

- I would suggest this more active title:

The actin nucleation promoting factor WASH facilitates clathrin-independent endocytosis of human papillomaviruses

- Please provide the abstract written in present tense throughout.

- We now use CRediT to specify the contributions of each author in the journal submission system. CRediT replaces the author contribution section. Please use the free text box to provide more detailed descriptions and do NOT provide your final manuscript text file with an author contributions section. See also our guide to authors:

<https://www.embopress.org/page/journal/14693178/authorguide#authorshippinguidelines>

- Please order the sections like this, using these names:

Title page - Abstract - Keywords - Introduction - Results - Discussion - Methods - Data availability section - Acknowledgements - Disclosure and Competing Interests Statement - References - Figure legends - Expanded View Figure legends

- I would suggest adding what is presently shown in the two Appendix Figures to the EV figures. We can have up to 5 EV figures. Then, we would not need the Appendix. Please do that and update the EV figure legends and all affected the callouts.

- Please remove the movie legends from the manuscript text file. Each legend needs to be provided in a readme.txt file. Then each movie should be ZIPped up with its corresponding legend file and uploaded separately - one ZIP folder per movie.

- Please check again that the number "n" for how many independent experiments were performed, their nature (biological versus technical replicates), the bars and error bars (e.g. SEM, SD) and the test used to calculate p-values is indicated in the respective figure legends. Please also check that all the p-values are explained in the legend, and that these fit to those shown in the figure. Please provide statistical testing where applicable. Please avoid the phrase 'independent experiment' but clearly state if these were biological or technical replicates. Please also indicate (e.g. with n.s.) if testing was performed, but the differences are not significant. In case $n=2$, please show the data as separate datapoints without error bars and statistics. See also:

<http://www.embopress.org/page/journal/14693178/authorguide#statisticalanalysis>

If $n < 5$, please show single datapoints for diagrams. Could statistics be provided also for the diagrams in 1B and EV1? Moreover, please add "n.s." to the diagrams. This is more comprehensible than stating 'no asterisk = not significant'). Moreover:

- Please note that the legend for figure EV3 A, B is missing in the manuscript. This needs to be rectified.

- Please note that the exact p values are not provided in the legends of figures 1C, 3A-D; 5A-G; EV3 A, C, E, F, G, H; appendix figures 1C

- Please note that the box plots need to be defined in terms of minima, maxima, centre, bounds of box and whiskers, and percentile in the legend of figure 2D

- Please note that information related to n is missing in the legend of figure EV3 H

- Please note that the error bars are not defined in the legend of figure EV3 H.

- Please add to each legend (main and EV figures, where applicable) a 'Data Information' section (or name the provided section like this) explaining the statistics used or providing information regarding replicates and scales. See:

- The Data Availability Section (DAS) is restricted to information regarding large primary datasets deposited at external databases (in this case the source data at BioStudies (including a direct link to the deposited data)).

- Please remove the Reagents & Tools Table from the manuscript text file and upload this as separate file, using the downloadable template (.doc) that can be found in our author guidelines (section 'Structured Methods'):

- There is a callout to a 'supplemental table 1' on page 36. Does this refer to the Reagents & Tools Table? Please check. Moreover, please name callouts for this table 'see Reagents & Tools Table'.

- Please provide a fully completed author checklist (including corr. author name, journal name and manuscript ID#).

- It seems that some of the images shown in Fig. 1A are also shown in Fig. 1B below the diagram. Please mention this reuse in the figure legend.

In addition, I would need from you uploaded separately:

- a short, two-sentence summary of the manuscript (not more than 35 words).

- two to four short (!) bullet points highlighting the key findings of your study (two lines each).

- a schematic summary figure as separate file that provides a sketch of the major findings (not a data image) in jpeg or tiff format (with the exact width of 550 pixels and a height of not more than 400 pixels) that can be used as a visual synopsis on our website.

Best,

Referee #2:

The authors have responded satisfactorily to my comments and have added new data that strengthen the manuscript, which I believe can now be published

Referee #3:

The authors adequately responded to my comments. This manuscript is nearly suitable for publication, but I suggest that the authors consider addressing a relatively new publication that regards related actin nucleation promotion factors in their discussion. <https://doi.org/10.3390/v17040542>

All editorial and formatting issues were resolved by the authors.

Prof. Mario Schelhaas
University of Muenster
Institute of Cellular Virology
von Esmerch Str 54
Münster 48149
Germany

Dear Prof. Schelhaas,

I am very pleased to accept your manuscript for publication in the next available issue of EMBO reports. Thank you for your contribution to our journal.

Yours sincerely,
